# Modelling of the urban concentrations of PM$_{2.5}$ on a high resolution for a period of 35 years, for the assessment of lifetime exposure and health effects

Jaakko Kukkonen[1], Leena Kangas[1], Mari Kauhaniemi[1], Mikhail Sofiev[1], Mia Aarnio[1], Jouni J.K. Jaakkola[2], Anu Kousa[3] and Ari Karppinen[1]

[1]Finnish Meteorological Institute, Erik Palmenin aukio 1, P.O. Box 503, FI-00101, Helsinki, Finland

[2]Center for Environmental and Respiratory Health Research, and Medical Research Center, P. O. Box 5000, FI-90014 University of Oulu, Finland

[3]Helsinki Region Environmental Services Authority, P.O. Box 100, FI-00066 HSY, Helsinki, Finland

*Correspondence to*: Jaakko Kukkonen (jaakko.kukkonen@fmi.fi)

**Abstract.** Reliable and self-consistent data on air quality is needed for an extensive period of time for conducting long-term, or even lifetime health impact assessments. We have modelled the urban scale concentrations of fine particulate matter (PM$_{2.5}$) in the Helsinki Metropolitan Area for a period of 35 years, from 1980 to 2014. The regional background concentrations were evaluated based on reanalyses of the atmospheric composition on global and European scales, using the SILAM model.

The high resolution urban computations included both the emissions originated from vehicular traffic (separately exhaust and suspension emissions) and those from small-scale combustion, and were conducted using the road network dispersion model CAR-FMI and the multiple source Gaussian dispersion model UDM-FMI. The modelled concentrations of PM$_{2.5}$ agreed fairly well with the measured data at a regional background station and at four urban measurement stations, during 1999

– 2014. The modelled concentration trends were also evaluated for earlier years, until 1988, using proxy analyses. There was no systematic deterioration of the agreement of predictions and data for earlier years (the 1980's and 1990's), compared with the results for more recent years (2000's and early 2010's). The local vehicular emissions were about five-fold higher in the 1980's, compared with the emissions during the latest considered years. The local small-scale combustion emissions

increased slightly over time. The highest urban concentrations of PM$_{2.5}$ occurred in the 1980's; these have since decreased to about to a half of the highest values. In general, regional background was the largest contribution in this area. Vehicular exhaust has been the most important local source, but the relative shares of both small-scale combustion and vehicular non-exhaust emissions have increased in time. The study has provided long-term, high-resolution concentration databases on regional and

urban scales that can be used for the assessment of health effects associated with air pollution.

# 1 Introduction

This study has been part of the multidisciplinary research programs entitled "The Influence of Air Pollution, Pollen and Ambient Temperature on Asthma and Allergies in Changing Climate, APTA" (http://www.oulu.fi/apta-en/) and "Understanding the link between Air pollution and Distribution of related Health Impacts and Welfare in the Nordic countries, NordicWelfAir" (http://projects.au.dk/nordicwelfair/). The overall objective of the APTA project was to assess the

influence of environmental exposures related to changing climate on asthma and allergies, and on cause-specific mortality. One of the aims of the NordicWelfAir project was to further understand the link between air pollution levels, the chemical composition of pollution and the related health effects.

A special focus of the present study was on health effects of air pollution, allergenic pollen and

extreme climatic factors, especially ambient temperatures. Asthma is globally the most common chronic disease in children (WHO, 2011) and the prevalence of asthma in adult populations varies from 5 to 25 %. Allergies are even more common, with prevalence of $30 - 40$ %. Accurate information on the long-term concentrations of air pollutants and pollen species on a high spatial and temporal resolution was required for conducting life-time health impacts assessments.

Only few studies have estimated the past levels air pollution over decades. However, some studies have been conducted, e.g., for the northern hemisphere (Xing et al., 2015) and for specific continents, regions or countries (Guerreiro et al., 2014, Colette et al., 2011, Cho et al., 2011, Astitha et al., 2017). All of these studies have addressed a relatively short time period, commonly from a few years up to two decades. The evaluations of air quality for urban areas over longer periods have been especially

rare (Gupta and Christopher, 2008, Wise and Comrie, 2005). Previous investigations have not addressed the modelling of spatially and temporally high resolution concentrations in cities or urban agglomerations over a period of several decades.

Xing et al. (2015) evaluated air quality trends during a 21-year period, 1990–2010, in the Northern Hemisphere, using both observations and modelling. They used the Community Multiscale Air

Quality (CMAQ) model driven by meteorology from Weather Research and Forecasting (WRF) simulations and the emission inventories from EDGAR. They concluded that the model successfully reproduced the observed decreasing trends in $SO_2$, $NO_2$, 8 h $O_3$ maxima, $SO_4^{2-}$ and elemental carbon (EC) in the US and Europe. However, the model also failed to reproduce part of the trends, such as that of $NO_3^-$ in the US, and some trends for China. Due to the coarse spatial resolution, the modelling

underestimated most concentrations in urban networks.

Guerreiro et al. (2014) presented an analysis of air quality in Europe from 2002 to 2011. The evaluation was based on ambient air measurements and data on anthropogenic emissions. They reported that the emissions of the main air pollutants in Europe declined in the considered period, resulting in improved air quality across the region, for some of the pollutants. The trends of PM10

were on average decreasing across Europe. Most of the registered significant trends of the measured PM10 concentrations were decreasing in monitoring stations that were situated mostly in France and Germany. However, the data at most of the stations did not show significant trends. In case of the $PM_{2.5}$ concentrations, most of the stations did not register significant trends in the period 2006 - 2011.

Colette et al. (2011) analyzed air quality trends in Europe during a decade, from 1998 to 2007, using both measurements and six regional and global chemistry-transport models. They presented the trends of $NO_2$, $O_3$ and $PM_{10}$, based on both the measured data and modelled results. However, this article was focused on the evaluation of the performance of the selected models.

Astitha et al. (2017) evaluated the trends of ozone concentrations over the contiguous United States from 1990 to 2010, using the coupled WRF-CMAQ system. They addressed various $O_3$ measures, such as, e.g., the daily maximum 8-hr ozone concentrations, during the warm seasons. During the earlier decade (from 1990 to 2000), the simulated and observed trends were not statistically significant. During the more recent decade, all observed trends were decreasing statistically significantly. Wise and Comrie (2005) analyzed meteorologically adjusted ozone and PM data from 1990 to 2003 for five major metropolitan areas in the Southwestern U.S. Long-term ozone trends indicated increasing concentrations over the considered decade; the long-term PM concentrations did not show any significant trends.

Gupta and Christopher (2008) analysed particulate matter concentrations in Birmingham, U.S., during a seven year period, 2002 – 2006, based on both surface and satellite measurements. They used the aerosol optical thickness (AOT) values determined by the Moderate Resolution Imaging Spectroradiometer (MODIS), and ground measurements of particulate matter mass. They concluded that the $PM_{2.5}$ mass concentrations over Northern Birmingham were decreased by about 23 %. However, the MODIS-Terra AOT data was available only about 50 % of the time, due to cloud cover and unfavorable surface conditions.

Cho et al. (2011) reviewed studies that addressed the concentrations and size distributions of ambient airborne particulate matter containing Pb, from 1980 to 2008. The maximum three month average airborne Pb concentrations in total suspended particulate matter decreased 92 % at 19 national sites in the U.S. during the considered period. However, the relevant size distribution data were very scarce.

The first systematic review of urban air quality in Finland was presented by Kukkonen et al. (1999). They considered the data from 42 Finnish cities and towns, in 1990-1993. The results showed that the national air quality guidelines were fairly often exceeded in the early 1990's in urban areas, most commonly for particulate matter. Some exceedances also occurred for $NO_2$ and CO, at sites with high traffic densities. However, the European Union air quality limit values were only exceeded at one monitoring station for all of the years considered. More recently, Anttila and Tuovinen (2010) evaluated the trends in the atmospheric concentrations of the main gaseous and particulate pollutants in Finland for the period of 1994–2007, based on measured data at both regional background and urban stations. They found that the concentrations of $SO_2$, CO and $NO_x$ declined considerably during the considered period, whereas the concentrations of O3 increased in the urban data. For $PM_{10}$, five of the studied 12 trends were decreasing at the urban locations.

The main empirical framework of the present study addressed the Helsinki Metropolitan Area (HMA), from 1980 to 2014. The Espoo Cohort Study (ECS), which is a major prospective, population-based epidemiologic study, formed the basis for investigating the health impacts, in addition to registry-based hospitalization and mortality data. The ECS includes detailed information on the lifetime history of the residences and work places of the cohort members, as well as extensive

information on health, environmental exposures and behavioral factors collected both prospectively and by linkage to available health and environmental information registries (Jaakkola et al., 2010, Paaso et al., 2014). However, for the assessment of lifetime environmental exposure, a database would first be required on the concentrations of the key pollutants, allergenic pollen species and meteorological factors over a time period of 35 years.

The first aim of this study was to provide an accurate and reliable high-resolution database on the urban scale concentrations of PM$_{2.5}$, to be used in the subsequent health impact assessments. This article presents the emission and dispersion modelling in the Helsinki Metropolitan Area, for three and a half decades. A new, comprehensive emission inventory has been presented for PM$_{2.5}$ originating from small-scale combustion in this area. We have also extended the urban emission

inventories regarding the exhaust and non-exhaust emissions of vehicular traffic, and those for small-scale combustion, for a multi-decadal period. The second aim was to evaluate the reliability and accuracy of the modelled concentrations, by comparing those with the corresponding available measured concentrations. The third aim was to present selected results regarding the long-term temporal evolution of the local scale emissions and concentrations of PM$_{2.5}$.


## 2 Materials and methods

### 2.1. Overview of the domain and the set-up of the computations

The Helsinki Metropolitan Area (HMA) and its surrounding regions are situated on a fairly flat coastal

area by the Baltic Sea. The climate is relatively milder, compared with most other areas in the same latitudes, caused by the warming effect of the Gulf Stream and the prevailing global atmospheric circulation. The HMA comprises four cities; Helsinki, Espoo, Vantaa and Kauniainen. The total population in the HMA is approximately 1.1 million in 2018, while the population of Helsinki is about 0.63 million inhabitants. The larger, so-called greater Helsinki region includes 1.5 million

inhabitants. The cities in this area, and the selected measurement stations have been presented in Fig. 1.

The previous studies in this area have found that the most important local source categories for both the concentrations of the PM mass fractions and the PM number concentrations have been vehicular traffic and small-scale combustion, with smaller contributions from shipping and harbour operations,

industrial sources, and aviation (Soares et al., 2014, Kukkonen et al., 2016 and Aarnio et al., 2016). The aim of this study was to evaluate the concentrations of PM$_{2.5}$ for a multi-decade period on a high temporal and spatial resolution. It was therefore necessary to simplify the computations, compared to corresponding assessments for shorter periods.

We have therefore addressed in this study explicitly only long-range transport, and local vehicular

traffic and small-scale combustion. The information on the temporal evolution of shipping, the technical specifications of ships, and the engines and emissions of ships is very scarce for the earlier parts of the target period.

Soares et al. (2014) evaluated the emissions from shipping and major stationary sources in this area for 2008. The emissions from ship traffic were modelled using the Ship Traffic Emissions Assessment Model (STEAM) presented by Jalkanen et al. (2016) and Johansson (2017). The geographical domain was selected to include all the major harbours in Helsinki. However, the secondary particulate matter from shipping was not included. They also could not include the influence of small-scale combustion. They found that over all receptor grid locations, long range transport, vehicular traffic and shipping contributed 86, 11 and 3 % to the $PM_{2.5}$ concentrations, respectively. However, considering in detail the high-resolution spatial concentrations distributions, Soares et al. (2014) found that the contribution of shipping to the total $PM_{2.5}$ concentrations was on the average higher than 20 % within a distance of one kilometre of the major harbours.

We have used a roadside dispersion model and a multiple source Gaussian model for evaluating the atmospheric dispersion on an urban scale; however, we have not modelled the dispersion in street canyons in detail. The computations were performed on a high temporal resolution, one hour, and on a high spatial resolution, ranging from 10 m to 200 m within the domain.

## 2.2 Meteorological pre-processing

Measured meteorological data was analyzed using the meteorological pre-processing model MPP-FMI that has been adapted for an urban environment (Karppinen et al., 2000a). The MPP-FMI model is based on the energy budget method of van Ulden and Holtslag (1985). The model utilises meteorological synoptic and sounding observations. The output of the model contains an hourly time series of the meteorological data that is needed for the local dispersion modelling, including temperature, wind speed, wind direction, Monin-Obukhov length, friction velocity and boundary layer height. The pre-processed meteorological parameters were evaluated for all hours during the target period. However, the predicted meteorological dataset is not spatially variable; i.e., the same meteorological parameters were used for the whole of the Helsinki Metropolitan Area in the dispersion calculations.

We used the synoptic weather observations at the station of Helsinki-Vantaa airport (18 km north of Helsinki city center) and at marine stations south of Helsinki (Katajaluoto, Isosaari and Harmaja), radiation observations at Helsinki-Vantaa, and sounding observations at the station of Jokioinen (90 km northwest of Helsinki) for the whole target period. We had to use a compilation of the wind data measured at several stations located south of Helsinki in marine environments, as the time series of data at any of these stations was not complete for the whole of the target period. All three marine stations were located on islands in the Gulf of Finland.

The synoptic data used in the model was based primarily on observations at Helsinki-Vantaa. At any specific time, the predicted wind values were, however, obtained as a combination of observational data at two stations. This procedure yields a better spatial representativity of the wind values for the considered domain, compared with using only the data at one station. The sounding data included observed vertical temperature and pressure profiles.

However, after the procedure described above, the time-series of all the relevant observations was still not totally complete as input for the meteorological pre-processor. Single (hourly) missing observations or short intervals of observations were therefore completed, by interpolating, or replacing the missing values by the corresponding values at each previous measured time instant. There was one longer period with missing data at the station of Helsinki-Vantaa (the month of April in 1986) that was replaced by (i) the corresponding synoptic data at a station in the city Turku, and (ii) the radiation data at the station of Jokioinen.

## 2.3 Evaluation of the emissions in the Helsinki Metropolitan Area

### 2.3.1 Evaluation of the traffic flows and vehicular emissions

The emission inventory included exhaust and suspension emissions originated from vehicular traffic for the network of roads and streets in the HMA.

### Evaluation of traffic flows and vehicular exhaust emissions in 2012

We have used the detailed geographical information on the line source network in the Helsinki Metropolitan Area that was provided by the Helsinki Region Transport. We also conducted a thorough checking of the accuracy of this data, based on independent information on the location of streets and roads reported by (i) Google Earth, (ii) National Land Survey of Finland and (iii) Helsinki Region Environmental Services Authority. The locations of the reported line sources were in a number of cases found to be substantially inaccurate and were therefore revised. The final number of line sources in the revised inventory was 26 536.

The traffic volumes and average travel speeds at each traffic link were computed using the EMME/2 transportation planning system for 2008 (Helsinki Region Transport, 2011). The traffic flow data included geographical information and vehicle mileage for the above mentioned line sources within the Helsinki Metropolitan Area, for three selected hours of each day. The traffic flow data also included average driving speeds for personal cars. The hourly traffic volumes were computed using a set of regression-based factors for the diurnal variation of hourly traffic flows, separately for weekdays, Saturdays and Sundays. These factors were estimated by the Helsinki Region Transport.

We also used the most recent data of the national calculation system for traffic exhaust emissions and energy consumption in Finland, called LIPASTO (Mäkelä and Auvinen, 2009; lipasto.vtt.fi). The system comprises two separate units, the emission inventory part and the unit emission database part. The LIPASTO system also includes data on mileages. The database contained city-level data for years 2012-2014, and the trends of emissions and mileages since 1980. The emission factors of vehicles are based on the guidelines of the Intergovernmental Panel on Climate Change for national greenhouse gas inventories in 2006, the Emission Inventory Guidebook of the European Monitoring and Evaluation Programme and the European Environment Agency, and on national laboratory measurements (http://lipasto.vtt.fi/liisa/liisa_menetelma.pdf). The LIPASTO system is similar, e.g.,

to the corresponding system that is used in the U.K. (AEA, 2009). The mileage trend data was used
to scale the 2008 traffic volume data to evaluate traffic flows in 2012.

Separate emission factors were used in this study for various classes of vehicle types, including
personal vehicles, vans, buses, and a combination of lorries and trucks. However, the emissions from
motorcycles and mopeds were not included. In case of personal vehicles, vans and lorries, we also
allowed for differing emissions for (i) the streets within the city centers and other highly populated
areas, and (ii) the roads in suburban areas and in the outskirts of the most populated urban areas.

### Evaluation of traffic flows and vehicular exhaust emissions for the whole target period

We have used the inventory of traffic flows and traffic exhaust emissions in 2012 in the Helsinki
Metropolitan Area as a starting point for evaluating the traffic flows and emissions for the whole
target period. This inventory contains also the detailed spatial distribution of the roads and streets,
their traffic flows and the evaluated emissions from all the main roads and streets in this area.

Evaluations of the annual values of the total vehicular emissions within the Helsinki Metropolitan
Area are contained in the LIPASTO system, since the early 1980's. These values are based on detailed
computations for the changing properties of the vehicle fleet, including factors, such as the
composition of the gasoline and diesel fuels, the various vehicle categories, the changing emission
standards for vehicles, the modes of driving (urban and highway conditions), occupancy of vehicles
(numbers of persons), etc. We have applied the trends of the total traffic flows and vehicular exhaust
emissions, based on the values in the LIPASTO system. The emissions for the road and street network
in the area were scaled for the other years using these annually and spatially averaged (for the whole
of the Helsinki Metropolitan Area) values.

The emission inventory for 2012 was scaled for the whole target period, by scaling the emission in
each road link. The scaling was done by multiplying the emissions in 2012 by the ratio of the total
exhaust emission values in the Helsinki Metropolitan Area in any selected year to that in 2012. The
total exhaust emissions in the area were extracted from the LIPASTO system. However, when
completing this procedure, one needs to assume that the spatial distribution of the traffic exhaust
emissions does not substantially change from year to year.

### Evaluation of the emissions of suspended dust for the target period

It was not possible to use detailed suspension emission models, such as the NORTRIP or FORE
models (Kauhaniemi et al., 2014) for such a long period. The input values for such models regarding
especially road sanding and the use of studded tyres were not available for the whole target period.
We have therefore used a simpler semi-empirical modelling approach for evaluating the emissions of
suspended dust.

First, we evaluated the fractions of suspended particle emissions based on model computations for a
couple of recent years, for which the detailed input information was available. Second, we evaluated

the evolution of these fractions for the whole period. We computed both the $PM_{2.5}$ concentrations originated from traffic exhausts, using the CAR-FMI model, and the suspended $PM_{2.5}$ concentrations, using the FORE model, for the whole of the Helsinki Metropolitan Area in 2008 and 2010. We have subsequently computed the monthly ratios of the concentrations, $C_{susp}/C_{exh}$, in which $C_{susp}$ is the suspended $PM_{2.5}$ concentration and $C_{exh}$ is the corresponding concentration originated from the traffic exhausts. These ratios were computed at four stations. The monthly average ratios were then used to evaluate the traffic suspension emissions. The annual average values of the ratio $C_{susp}/C_{exh}$ for $PM_{2.5}$ were 0.33 and 0.36 in 2008 and in 2010, respectively.

We evaluated the deviations of the above described simpler semi-empirical method, in comparison with the computations by the FORE model for three years, 2012-2014. The predicted annual average suspended $PM_{2.5}$ concentrations at four measurement stations (Mannerheimintie, Kallio2, Leppävaara4 and Tikkurila3) computed with the semi-empirical method ranged from 67 to 98 % of the predictions by the FORE model. The monthly averaged variation of the suspended concentrations was slightly more moderate, using the simpler model.

The computations using the FORE model allow for the suspended dust originated from sanding and the wear of the road surfaces, including especially the wear caused by the studded tyres. However, the emissions of tyre, brake and clutch wear are not explicitly included in the FORE model. Kupiainen et al. (2015) have estimated that the particulate matter emissions for tyre, brake and pavement wear in the Helsinki Metropolitan Area were less than a third of the total vehicular non-exhaust emissions. The model computations of the present study include the rest of the non-exhaust emissions; these contain approximately two thirds of the whole, and in addition a substantial part of the remaining third of the non-exhaust emissions (i.e., the wear of road surfaces due to studded tyres). The omitted fraction of the predicted vehicular non-exhaust emissions can therefore be estimated to be clearly smaller than a third.

The estimation of suspended dust emissions for the other years was done using the emissions and traffic volumes, obtained from the LIPASTO system. The ratio of vehicular suspension and exhaust emissions during year y is by definition

$$R(y) = \frac{e_{susp}(y)}{e_{exh}(y)} \qquad (1),$$

where $e_{susp}$ and $e_{exh}$ are the suspension and exhaust emissions, respectively.

For simplicity, we assume that the total emission from vehicular suspension sources is directly proportional to the total traffic volume, both evaluated within the considered domain for each year. The annual vehicular suspension emission in the domain is also dependent on other factors, especially on the fraction of studded tyres in cars, characteristic traffic speeds and the types of street surfaces used. However, sufficiently detailed information on these factors for the whole of the target period was not available.

Equation (1) can therefore be written as

$$R(y) = \frac{\frac{V(y)}{V(T)} e_{susp}(T)}{e_{exh}(y)} \qquad (2),$$

Where V(y) and V(T) are the traffic volumes during the years y and any other year T. T can be selected to be the target year, 2012.

Using equation (1) to rewrite $e_{susp}$(T) yields

$$R(y) = \frac{V(y)}{V(T)} R(T) \frac{e_{exh}(T)}{e_{exh}(y)} \qquad (3).$$

In equation (3), the ratios of the traffic volumes V(y)/V(2012) and those of the exhaust emissions $e_{susp}$(2012)/$e_{exh}$(y) can be computed. The ratio of vehicular suspension and exhaust emissions during the target year R(2012) is also known, based on the above mentioned evaluation.

The vehicular suspension emissions can therefore be computed for any historical year (y) based on
equation (3), if the exhaust emissions and the total traffic volume during that year are known. We have used this approach for evaluating the suspension emissions for all the other years, except for 2012.

### 2.3.2 Evaluation of the emissions from small-scale combustion


**A novel emission inventory for the small-scale combustion of wood in 2014**

A novel emission inventory was compiled for 2014 for the following pollutants: particles ($PM_1$, $PM_{2.5}$ and $PM_{10}$), nitrogen oxides (NOx), non-methane volatile organic compounds (NMVOC), carbon monoxide (CO), black carbon (BC) and benzo(a)pyrene (BaP). For a more detailed description of this
inventory, the reader is referred to Hellen et al. (2016) and Kaski et al. (2016).

The procedures and habits of combustion, and the amount of combusted wood have been estimated by a questionnaire (Kaski et al., 2016a). Wood combustion has been commonly used both as a supplementary heating method and as a fuel in the stoves in saunas. In total, wood combustion has been used as either a primary or secondary heating method in approximately 90 % of the detached
and semidetached houses in the area. However, wood combustion has seldom been used as a primary heating method, only in approximately 2 % of the houses.

The small-scale combustion that is not using oil or gas in this domain uses almost solely wood as a fuel. The share of pellet use as a fuel has been negligible. The average amount of wood burned per house was annually 1.52 solid cubic meters. Most of the wood was used in heat-storing masonry

heaters (0.72 solid-m$^3$/house) and sauna stoves (0.31 solid-m$^3$/house); only a minor amount was burned in heating boilers (0.09 solid-m$^3$/house) and other devices.

## The emission factors and total emissions from small-scale wood combustion

The emission factors of PM$_{2.5}$ were evaluated for a unit of energy contained in the wood fuels (mg/MJ). The evaluations of emission factors were mainly based on nationally conducted laboratory

measurements (Kaski et al., 2016).

The emission factors were first separately analyzed for 12 considered heating units for the year 2014. The applied values were (all in units of mg/MJ) masonry heater 125, convection fireplace 49, open fireplace with convection 125, open fireplace without fire doors 638, a combination of fireplace and baking oven 125, a combination of stove and baking oven 53, wood-burning stove 53, baking oven

48, wood-burning oven 93, wood-burning sauna heater 470, heating boiler 248, and other fireplaces (on average) 470.

Considering also the frequencies of the use of these units within the whole of the Helsinki Metropolitan Area in 2014, it was computed that most of the emissions originated from wood combustion were caused by wood-burning sauna heaters (77 tons) and masonry heaters (48 tons).

According to these computations, these two heating categories were responsible for 71% of the total emissions of PM$_{2.5}$ from wood burning in the area (175 tons) in 2014.

## The temporal evolution of the emissions from small-scale wood combustion

There are numerous factors that affect the long-term (decadal) temporal evolution of the emissions from small-scale wood combustion. For estimating the amount of wood combustion, it is essential to

know (i) the numbers of detached and semidetached houses, (ii) the amounts of firewood used, (iii) the shares of primary heating sources, (iv) the numbers of boilers and sauna stoves, and (v) the changes of technology and fuels during the considered period. We have allowed for the influence of the above mentioned factors (i) – (iv), for the whole of the considered period. Regarding item (iv), we have also allowed for the temporal changes of the usage of the main categories of technologies

and heating devices.

However, there are also other factors, which we evaluated to be relatively less influential; these were not taken into account in detail. These include (i) the detailed composition of wood fuels, including their humidity and the fractions of various tree species, and (ii) the changes of the habits and procedures of combustion. The former factor refers to the ways, by which the wood fuels are

collected, pre-processed and stored, and the latter how these are ignited and combusted.

We used the wood combustion emissions evaluated for 2014 as a starting point. The total emission values for the other years were estimated by using previous available emission surveys and various statistical data. In addition to the above mentioned survey for 2014, the Helsinki Region Environmental Services Authority has estimated the amount of wood combustion in the area for 2002.

This survey was conducted using the results of the previous corresponding studies in this domain, and various statistical data. In addition, the emissions were evaluated for 2007 by a questionnaire.

The numbers of detached and semidetached houses were extracted from the Helsinki Region Trends statistical database, and from the statistical data reported by the urban municipalities in this area. The estimate on the amount of firewood used is based both on the above mentioned questionnaire survey for the year 2014, and on a survey by the Forest Research Institute in Finland for the years 1992 and 1993, and 2000 - 2001. The shares of primary heating sources are based on statistical data and the above-mentioned surveys by the Helsinki Region Environmental Services Authority.

The historical fractions of sauna stoves in detached and semidetached houses during previous years was evaluated partly based on Luoma (1997). The estimates in this inventory are based on a limited amount of data, regarding 300 detached and semidetached houses. According to Luoma (1997), there was a sauna stove in approximately 50 % of the detached and semi-detached houses in this area in 1980. According to more recent statistics by the Helsinki Region Environmental Services Authority, this share has decreased to 28 % in 2013. The amounts of firewood used for sauna heating were evaluated based on the survey by the Helsinki Region Environmental Services Authority. The amount of heating boilers was analyzed based on Luoma (1997) and the surveys by the Helsinki Region Environmental Services Authority. For boilers, the number of heating days per year was also taken into account.

We have evaluated separately the temporal trends of the frequencies of three fireplace categories, i.e., sauna stoves, boilers and all other devices. Regarding sauna stoves, the technologies have changed very slowly. Only a low fraction of sauna stoves are currently based on more modern technologies with relatively lower emissions (Savolahti et al, 2016). There is also information on more modern, so-called low-emission masonry heaters. These devices have been available since 2000; and their number has been assumed to have increased linearly since that time. Their share was 7.1 % of all masonry heaters in 2010 (Kaski et al., 2016). However, there is not enough information to take into account the changes of technological developments for the various heating devices during the 20th century.

## 2.4 Atmospheric dispersion modelling

### 2.4.1 Urban scale dispersion modelling

The urban scale dispersion of vehicular emissions was evaluated with the CAR-FMI model (Contaminants in the Air from a Road – Finnish Meteorological Institute; e.g., Kukkonen et al., 2001). The model computes an hourly time-series of the pollutant dispersion from a network of line sources. The dispersion equation is based on a semi-analytical solution of the Gaussian diffusion equation for a finite line source. The dispersion parameters are modelled as a function of the Monin-Obukhov length, the friction velocity and the mixing height. Traffic-originated turbulence is modelled with a semi-empirical treatment.

The modelling system containing the CAR-FMI model has been evaluated against the measured data of urban measurement networks for gaseous pollutants (e.g., Karppinen et al., 2000c and Kousa et al., 2001) and for $PM_{2.5}$, $PM_{10}$ and particle number concentrations in the Helsinki Metropolitan Area

(Kauhaniemi et al., 2008, Aarnio et al., 2016 and Kukkonen et al., 2016). The model has also been evaluated both against gaseous and particulate pollutant measurements in London (Sokhi et al., 2008, and Singh et al., 2013 and Srimath et al., 2016) and in Birmingham, U.K. (Srimath et al., 2016). The performance of the CAR-FMI model has also been evaluated for gaseous pollutants against the results of field measurement campaigns and inter-compared with other models (Kukkonen et al., 2001, Oettl et al., 2001, Levitin et al., 2005, Srimath et al., 2016).

The dispersion from small-scale combustion sources was evaluated using the UDM-FMI model (Urban Dispersion Model – Finnish Meteorological Institute; Karppinen et al., 2000b). The model is based on multiple source Gaussian plume equations for various stationary source categories (point, area and volume sources). The modelling system including the UDM-FMI and CAR-FMI models has been evaluated against the measured data of urban measurement networks for gaseous pollutants (e.g., Karppinen et al., 2000b and Kousa et al., 2001) and for $PM_{2.5}$ (Kauhaniemi et al., 2008).

In this study, the emissions from domestic wood combustion were uniformly distributed in area sources of size 100 m x 100 m. The altitude of the releases for domestic wood combustion was assumed to be equal to 7.5 m, including the initial plume rise. This altitude value is based on the average heights of the detached and semidetached houses and their chimneys within the study domain, and an estimated plume rise. The dispersion was evaluated separately for three different emission source categories: sauna stoves, boilers, and other fireplaces. The diurnal, weekly, and monthly variations within the emission inventory were applied for each source category.

Both for the dispersion of vehicular and small-scale combustion pollutants, $PM_{2.5}$ was treated as inert, i.e., no chemical or physical transformation was assumed to take place within the urban time scales.

### 2.4.2 Regional scale dispersion modelling

The air quality measurements that are representative for the regional background of this area have been conducted at the station of Luukki. However, these measurements were started only after more than a half of the target period had passed, in 1999 and 2004 for $PM_{10}$ and $PM_{2.5}$, respectively. It was therefore not feasible to use the measured regional background concentrations for the whole period. Instead, we used modelled regional background concentrations, which constituted a self-consistent time series for the whole target period.

SILAM is a global-to-meso-scale dispersion model developed for evaluating atmospheric composition and air quality, and for emergency decision support applications, as well as for solving inverse dispersion problems (e.g., Sofiev et al, 2006). The model incorporates both Eulerian and Lagrangian transport routines. There are eight chemico-physical transformation modules (basic acid chemistry and secondary aerosol formation, ozone formation in the troposphere and the stratosphere, radioactive decay, aerosol dynamics in the air and transformations of pollen) and modules for three- and four-dimensional variational data assimilation. The source term descriptions of the SILAM model include point- and area- source inventories, sea salt, wind-blown dust, natural pollen, natural volatile organic compounds, nuclear explosion, as well as interfaces to the ship emission system STEAM and the fire information system IS4FIRES.( http://silam.fmi.fi/).

In the present study, we have conducted reanalyses of the atmospheric composition and air quality for the period 1980-2014 on global, European, and northern European scales, using the SILAM model (version 5.5). The simulations were conducted in three spatial domains, i.e., global (the troposphere and the stratosphere), European (troposphere) and Northern European ones (troposphere). However, as a combination of meteorological inputs was used for computing the concentrations in the Northern

European domain, these computations did not provide a completely homogeneous time series. We therefore used the corresponding results for the European domain as boundary conditions for the urban scale computations.

An overview of the global and regional computations is presented in the following. For a more detailed description, the reader is referred to Sofiev et al. (2018).

The global reanalysis was done using a 1.44°×1.44° longitude-latitude resolution, for both the troposphere and the stratosphere, for the period 1980-2014. The meteorological data was the ERA-Interim archive taken at full resolution, 0.72°×0.72°, and the vertical coverage included 61 hybrid levels (Dee et al., 2011; Simmons et al., 2010). For the global and European reanalysis, we have used the following emission data: MACCity and EDGAR (EDGAR 2014) for the anthropogenic PM

emissions (Janssens-Maenhout et al., 2017), ACCMIP for wild-land fires (Lamarque et al., 2010, Granier et al., 2011), MEGAN for biogenic emissions (Guenther et al., 2006), and GEIA for the emissions from lightning (Price et al, 1997) and aircrafts. The emissions of sea salt (Sofiev et al., 2011), wind-blown dust, and biogenic VOC's (for the European and Northern European domains; Poupkou et al., 2010) were evaluated using the embedded modules in SILAM.

The model output on global scale contains hourly surface and column-integrated concentration, and column aerosol optical depth (AOD), as three-hourly values at 18 model levels. There were 50 chemical species in the model output.

The European reanalysis was done using a longitude-latitude resolution of 0.5°×0.5°, for the troposphere, for the period 1980-2014. The meteorological input data for the dispersion modelling

was the ERA-Interim for the global and European domains (Dee et al., 2011, Simmons et al., 2010). The zoomed re-analysis for Northern Europe at a resolution of 0.1°×0.1° used BaltAn meteorological reanalysis by the Estonian Meteorological Institute. As the values included in the BaltAn reanalysis extend only up to the year 2005, this dataset was expanded by using the data from the ECMWF operational archives for 2006-2014.

The model output contained hourly three-dimensional surface, and column-integrated concentration and column AOD, at 13 modelled vertical levels. The model output included 40 chemical species.

In order to obtain the time series of background concentrations, we first selected four grid points of the SILAM computations that were closest to the HMA, but outside the urban domain. We then computed an hourly average of the concentration values at these four locations, and used that value

as the regional background for all the chemical components of particulate matter, except for mineral dust. This method is less sensitive to potential occasional local influence of the metropolitan area on the regional background concentrations, compared with using the predicted value only at one specific grid point. In case of mineral dust, we used the lowest hourly value within the four selected points.

The latter procedure was adopted to avoid the potential double counting of occasional releases of dust originating from the considered urban area.

## 3. Results and discussion

First, the predicted concentrations are evaluated against the measured values. Second, selected predicted results are presented, especially on the temporal evolution of the local emissions and urban concentrations. Third, the concentrations are analyzed in terms of the contributions originated from the various pollution source categories.

We also analyzed the contributions of shipping and harbor operations on the concentrations of PM$_{2.5}$. These results have been presented in Annex A. The contribution of shipping and harbours to the total PM$_{2.5}$ concentrations varies from 10-20 % in the close vicinity of the three major harbours to a negligible contribution in the northern and western parts of the area.

### 3.1 Evaluation of the predicted results against the measured concentrations

### 3.1.1 Statistical parameters for the comparison of predictions and measurements

Four statistical parameters were computed: the index of agreement (IA), the factor-of–two (F2), the coefficient of determination (R$^2$) and the fractional bias (FB). The parameters IA, R$^2$ and F2 are measures of the correlation of the predicted and observed time series of concentrations, whereas FB is a measure of the agreement of the predicted and observed mean concentrations.

The original form of the index of agreement (IA) is calculated as (Wilmott, 1981)

$$IA = 1 - \left[ \frac{\sum_{i=1}^{n}(x_i - y_i)^2}{\sum_{i=1}^{n}\left[|x_i^1| + |y_i^1|\right]^2} \right]$$

where n is the number of data points, and x and y refer to the predicted and measured pollutant concentrations, respectively. The symbols with the superscript '1' are defined as:

$$x_i^1 = x_i - \overline{y} \; ; \text{ and}$$

$$y_i^1 = y_i - \overline{y} \quad ,$$

in which the overbar refers to an average value. Also other forms of IA have been presented in the literature.

The IA, as defined above, ranges from 0.0 (theoretical minimum) to 1.0 (perfect agreement). However, even for a random predicted distribution, the IA value will be larger than the theoretical minimum. Karppinen et al. (2000b) found that for a random predicted distribution varying from zero to twice the measured average concentration, the IA value was approximately 0.40.

F2 is a measure of how many predictions are within a factor of two compared with the observations. Fractional bias (FB) ranges from - 2.0 to + 2.0 for extreme under- and over-prediction, respectively. The FB values equal to - 0.67 and + 0.67 are equivalent to under-and over-prediction by a factor of two, respectively.

### 3.1.2 Statistical analysis of the agreement of the predicted and measured concentrations

We have considered all the measurement data that have been available for the five measurement stations considered. For the stations of Kallio2, Luukki, Mannerheimintie, Tikkurila3 and Leppävaara4, the measured concentrations of $PM_{2.5}$ were available since 1999, 2004, 2005, 2009 and 2010, respectively.

### Regional background concentrations at one station

We first compared the predictions of the SILAM model for the regional background at the station of Luukki with the available measurements. The above mentioned statistical parameters were computed for both the hourly and daily concentrations of $PM_{2.5}$, separately for each year. The results for the daily concentrations are presented in Table 1. The corresponding scatter plots of the predicted and measured concentrations have been presented in Figs. 2a-k.

The IA values ranged from 0.60 to 0.73, and the F2 values range from 54 to 64 %. These IA values indicate that the temporal variation of the predicted daily $PM_{2.5}$ concentrations agrees reasonably well with the observed data. There are also no evident temporal trends in the statistical parameters during this period, i.e., the performance of modelling does not deteriorate for the earlier years. The fractional bias values in Table 1 indicate a slight under-prediction in most cases, the bias values ranging from − 14 to + 11 %.

The regional background was represented by the computations of the SILAM model on a European scale, with a spatial resolution of 0.5 degrees. Such a resolution may not be sufficient for achieving a high short-term (daily or hourly) temporal correlation of the measured and predicted time-series. The coastal meteorological effects could also cause inaccuracies on the shorter term correlations. In this study, the daily correlations were reasonable; however, the agreement of the long-term mean (e.g. monthly or annual) values of measured and predicted data was good.

### Urban concentrations at four stations

We have selected the data of four urban measurement stations for the evaluation of the predicted results. These stations and their classifications are as follows: Kallio2, urban background, Mannerheimintie, urban traffic, Leppävaara4, suburban traffic and Tikkurila3, suburban traffic. The stations of Kallio2 and Mannerheimintie are situated in central Helsinki, the station of Leppävaara4 is located in the city of Espoo and the station of Tikkurila3 in the city of Vantaa. The stations of Leppävaara and Tikkurila are also influenced by small-scale combustion from the surrounding residential areas. For simplicity, these stations are referred to in the following as KAL, MAN, LEP and TIK.

The above mentioned statistical parameters were computed for both the hourly and daily concentrations of $PM_{2.5}$ at the four measurement stations, separately for each year. The results for the daily concentrations are presented in Table 2. The corresponding scatter plots of the predicted and measured concentrations for one of the stations (KAL) have been presented in Figs. 3a-p. The time series of the measured and predicted annual average concentrations have also been presented, in Annex B.

The daily IA values range from 0.64 to 0.81, from 0.61 to 0.84, from 0.70 to 0.77 and from 0.71 to 0.83 at the stations of KAL, MAN, LEP and TIK, respectively. These fairly high or high IA values indicate that the temporal variation of the predicted daily $PM_{2.5}$ concentrations agrees well or fairly well with the observed data. The other considered statistical values can also be considered to indicate a good or fairly good agreement. In particular, there is no deterioration of the agreement of predictions and data for earlier years, compared with the corresponding results for more recent years. The scatter plots presented in Figs. 3a-p do not show any systematic under- or over-prediction of pollutant concentrations for the majority of the predicted concentrations.

The scatter plot of the predicted and measured annual averages is illustrated in Fig. 4. The results show that the differences of the modelled and measured annual average concentrations are less than 20 % in all the cases, except for two (Fig. 4, MAN 2010 and 2011). For three stations (KAL, LEP and TIK), there are no notable biases; however, for the station of MAN, there is a systematic under-prediction. The under-prediction could be caused by the reduced dilution caused by buildings, as this site is located in the center area of the city. The MAN site is also close to a junction of two heavily trafficked streets, and there is frequently a congestion of traffic. In particular, a substantial road re-construction work was in progress along this street in the vicinity of the site for part of the time in 2010 – 2011; this could possibly have caused relatively higher measured concentrations.

The indices of agreement and the fractional biases for both hourly and daily values have been presented graphically in Figs. 5a-b. As expected, the IA values are lower for the hourly values, compared with the daily values, caused by the larger scatter of hourly values, compared with the daily ones.

The agreement of the temporal variations of the predictions and measured data (as measured, e.g., by the index of agreement) is on the average slightly better for the urban stations, compared with that at the regional background station of Luukki. This is probably caused by the fact that the modelling of the temporal variations of the emissions and dispersion from local sources is more accurate, compared with those on a regional scale.

The model performance values can be compared with corresponding model evaluation studies in other cities. For instance, Singh et al. (2014) reported the analysis of modelled and measured fine particulate matter ($PM_{2.5}$) concentrations within London for 2008, evaluated using the OSCAR Air Quality Assessment System. Measured concentration values were used to model the temporal variation of the regional background concentrations. The evaluation was conducted with measurements from the London air quality network. For the predicted and measured hourly time

series of concentrations at 18 sites in London, the index of agreement of all stations was 0.86.

The corresponding IA values for the hourly $PM_{2.5}$ concentrations at the above mentioned four urban stations in the present study ranged from 0.60 to 0.73, from 0.55 to 0.78, from 0.64 to 0.69 and from 0.66 to 0.76 at the stations of KAL, MAN, LEP and TIK, respectively. The IA values in the present study were therefore slightly lower, although comparable with those found by Singh et al. (2014).

However, the somewhat better performance can also be partly caused by the use of measured concentration values in evaluating the temporal variation of the regional background by Singh et al. (2014).

## 3.2 The trends of urban emissions and concentrations


The predicted emissions originated from vehicular traffic and small-scale combustion in the Helsinki Metropolitan Area are presented in Fig. 6. The vehicular emissions were clearly higher in the 1980's and early 1990's, approximately five-fold, compared with the corresponding values during the latest considered years. The vehicular emissions were highest from 1986 to 1988. During the early 1980's,

there was a continuous increase of road traffic exhaust emissions both at the Helsinki Metropolitan Area and in Finland (Mäkelä and Auvinen, 2009).

The decreasing trend of vehicular exhaust emissions since the late 1980's has been achieved due to an increased use of catalytic converters and improved engine technology (Mäkelä and Auvinen, 2009). The relative share of suspension emissions (compared with the total vehicular emissions) has

been clearly higher during the most recent years.

During the considered period, the total mileage (as kilometers travelled per annum) in this area has grown continuously, except for a slight plateau during the early 1990's, from approximately 28 000 to 58 000 millions of km/a (Mäkelä and Auvinen, 2009). The mileage has also been projected to continue increasing in the foreseeable future. Although the mileage has almost doubled, at the same

time the vehicular emissions have decreased approximately to a fifth of their original value.

There is a slight increasing trend in the emissions from small-scale combustion. During recent years, the mass-based emissions from small-scale combustion have been only slightly smaller than those from vehicular traffic.

The temporal evolution of the modelled annual average $PM_{2.5}$ concentrations has been presented in

Fig. 7 at four stations in the HMA, together with the modelled regional background concentrations. According to these computations, the highest concentrations have occurred in the early and middle years of the 1980's. In 2014, the concentrations have decreased to about a half of the highest values

during the considered period. This has been caused both by the decreasing regional background and the decreasing local contributions.

The regional background is clearly the largest fraction of the total concentrations. The concentrations were systematically highest at the stations with the most dense traffic in the vicinity of the station (MAN and LEP), and relatively lowest at the urban background station of Kallio. However, during the last decade, there has been only a very slight decreasing trend.

The measured time series of concentrations of $PM_{2.5}$ extend from the present to the years from 1999
to 2010, depending on the station (cf. Table 2). However, there are measurements of the concentrations of $PM_{10}$ and TSP (total suspended particles) for substantially longer periods. These concentrations can be used as proxy variables for the measured concentrations of $PM_{2.5}$. The accuracy of the modelled trends of the concentrations of $PM_{2.5}$ can therefore be indirectly evaluated for earlier years using such a procedure. Such an analysis has been presented in Annex C.

The trends of the concentrations, as evaluated using the proxy values based on the measured $PM_{10}$ concentrations, agreed well at four stations, and fairly well at one station, with the modelled long-term evolution of concentrations. This adds some confidence that the modelled trends are fairly accurate.

**3.3 The contributions originated from various source categories**

The modelled urban source contributions (vehicular exhaust, vehicular suspension and small-scale combustion) have been presented in Figs. 8a-d. For clarity, the regional background values have not been presented in this figure. The local contributions range from 0.5 to 4.5 $\mu g/m^3$. Vehicular exhaust
has commonly been the largest urban contribution. However, at the stations that are most directly influenced by small-scale combustion (TIK and LEP), that contribution is also comparable with that of vehicular sources, during the most recent years.

**4. Conclusions**

We have modelled the concentrations of fine particulate matter globally, in Europe and in the Helsinki Metropolitan Area for a period of 35 years, from 1980 to 2014. The present study addresses the results in case of $PM_{2.5}$ for the urban area. Results on urban concentrations have not previously been presented in the literature on such a high spatial (tens or a couple of hundreds of meters) and temporal
(hourly) resolution for several decades. Examination of such historical results on the local emissions and concentrations also allows a wider perspective in view of the future evolution of air quality and its effects. These long-term results and trends were also thoroughly evaluated against the measured data at one regional background station and at four urban stations.

The results were subsequently post-processed to provide for the life-time concentrations of $PM_{2.5}$ for
each of the members of the Espoo Cohort Study. The cohort population included a random sample of

all children of the city of Espoo, born between 1984 and 1989, in total 2568 children (Jaakkola et al. 2010, Paaso et al. 2014). There is information on the complete history of the places of residence for the members of the cohort, as well as on their workplaces in adult age. We have therefore allowed for the changes of their life-time exposure, caused by the changes of the residences, separately for each cohort member. This analysis has enabled us to assess the health effects of life-time, time-specific, both long- and short-term exposures.

Clearly, there are numerous challenges in modelling accurately the emissions and atmospheric dispersion for such an extended period. Various factors affecting the relevant emissions, the emission coefficients and the absolute amounts of emissions are known less accurately for the earlier period, i.e., especially for the 1980's and also for the 1990's. This is the case regarding both the global and European scale emission inventories, and the urban scale inventories.

We have examined the available quantitative information on the changes of the street and road network in the considered area, but that information was not sufficiently detailed to be included in the emission modelling. The fractions and types of studded tires used in vehicles in the 1980's and 1990's were not known sufficiently accurately to be included in the modelling. It was also not possible to use more detailed models, such as those for evaluating the vehicular suspension emissions, and those for street canyon dispersion, due to missing input data (in the former case) and practical limitations on the amount of computations (in the latter case). The regional scale computations rely on the data of several previously compiled emission inventories; the accuracy of these datasets could potentially be worse for earlier decades.

A novel, detailed emission inventory for small-scale wood combustion was compiled in this metropolitan area for 2014 for a wide range of pollutants. The emission values provided by this inventory were also extended for a multi-decadal period. There are numerous factors that affect the decadal temporal evolution of the emissions from small-scale wood combustion. These include (i) the numbers of detached and semidetached houses, (ii) the amounts of firewood used, (iii) the shares of heating sources, boilers and stoves, and (iv) the changes of technology and fuels. We have evaluated the temporal evolution of the above mentioned factors (i) − (iii), and modelled their influence on the emissions from wood combustion, for the considered period. We have also allowed for the temporal changes of the usage of the main categories of technologies and heating devices. However, there were also other factors, which were not taken into account in detail. These include (i) the possible changes of the detailed composition of wood fuels and (ii) the changes of the habits and procedures of combustion.

The results of this study include a novel quantification of the emissions from two main local source categories, i.e., local vehicular traffic and small-scale wood combustion, for a multi-decadal period. The local vehicular emissions of $PM_{2.5}$ were about five-fold higher in the 1980's, compared with the emissions during the latest considered years. This substantial decrease of vehicular emissions has been achieved, due to an increased use of catalytic converters and improved engine technologies. During the same period, the total mileage (as kilometers driven per annum) in the area has almost doubled. However, the emissions originated from small-scale combustion have slightly increased in time.

The modelled concentrations of $PM_{2.5}$ agreed fairly well with the measured data at four measurement stations, during 1999 – 2014. These stations represented regional background, urban background, urban traffic and suburban traffic environments. Two of these stations were also influenced by small-scale combustion from the surrounding residential areas. However, we did not find any systematic deterioration of the agreement of predictions and data for earlier years (the 1980's and 1990's), compared with the corresponding results for more recent years (2000's and early 2010's). This gives more confidence that the emission inventories used for the earlier years were not substantially more inaccurate, compared with those for the more recent years.

The measured concentrations of $PM_{2.5}$ were slightly under-predicted for most years at the regional background station. However, these were not systematically under- or over-predicted for most of the urban stations and years. There was a systematic under-prediction of concentrations for one station, viz. Mannerheimintie in central Helsinki. Part of this under-prediction was probably caused by the reduced dilution caused by buildings and the increased vehicular emissions resulting from the frequently congested traffic conditions. The road re-construction works in the vicinity of this station for part of the time during two years could also have caused a slight additional contribution to the measured values.

We also evaluated the accuracy of the modelled trends of concentrations of $PM_{2.5}$, by using the measured values of the concentrations of $PM_{10}$ and TSP (total suspended particles) as proxy variables. Using such an analysis, the model results can be evaluated for a substantially longer period than would be possible using only the $PM_{2.5}$ measurements. In particular, the trends of the concentrations using $PM_{10}$ as a proxy agreed well at four stations, and fairly well at one station, with the modelled long-term evolution of concentrations. This adds more confidence that the modelled trends are fairly accurate.

The highest concentrations of $PM_{2.5}$ have occurred in the early and middle years of the 1980's. The urban concentrations of $PM_{2.5}$ have decreased to about a half of their highest values during the considered period. However, during the last decade, there has been only a very slightly decreasing trend. The regional background constitutes clearly the largest fraction of the total concentrations. Vehicular exhausts have commonly been the most important local source, but the relative share of small-scale combustion has continuously increased.

The study has provided high-resolution concentration databases on global, European and urban scales. The global and regional scale data is available to be used for similar smaller domain, high-resolution assessments worldwide, on request to the authors. The global or regional scale data can be used as boundary conditions for any selected location, for the time period from 1980 to 2014. Clearly, this could in many cases be much more cost-effective than starting from scratch to conduct such large scale emission and dispersion computations. The urban scale data has already been used for the assessment of individual-level life-time exposure to air pollution and its effects, especially with respect to the risk of developing asthma during the first three decades of life in the Espoo Cohort Study (ECS).

**5. Acknowledgements**

This study has been part of the following research projects: "The Influence of Air Pollution, Pollen and Ambient Temperature on Asthma and Allergies in Changing Climate, APTA" and "Global health risks related to atmospheric composition and weather, GLORIA", both funded by the Academy of Finland, grants #266214, #267675, #310372 and #310373, "Understanding the link between Air pollution and Distribution of related Health Impacts and Welfare in the Nordic countries", project #75007 (NordicWelfAir), funded by Nordforsk, and "Environmental impact assessment of airborne particulate matter: the effects of abatement and management strategies" (BATMAN), grant #286719, funded by the Academy of Finland. The funding from these two agencies is gratefully acknowledged. We also wish to thank Prof. Nicolas Moussiopoulos for his useful comments as a reviewer, and Helsinki Region Transport (HSL) for the traffic flow data.

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

As expected, the contribution of shipping and harbours is strongly focused in the vicinity of the major harbours. The distribution is dispersed especially around the central Helsinki area, and in the south-eastern coastal regions of the metropolitan area. The detailed comparison of the results in Figs. A1 a and b shows that the contribution of shipping and harbours to the total $PM_{2.5}$ concentrations varies from 10-20 % in the close vicinity of the three major harbours to a negligible contribution in the

northern and western parts of the area.

Annex B. The time series of the observed and predicted annual average concentrations of PM$_{2.5}$.

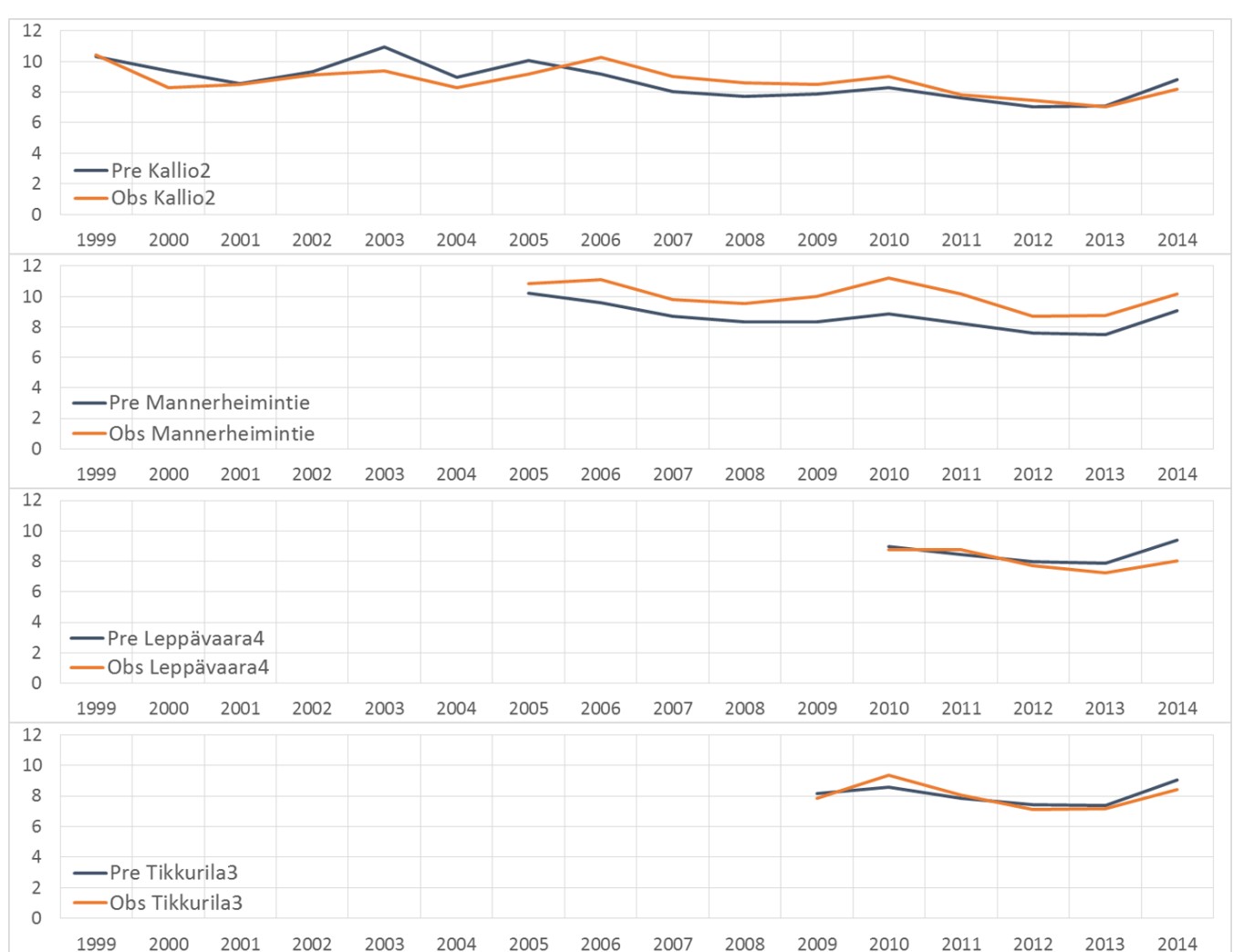

Fig. B1a-d. The time series of the observed and predicted annual average concentrations of PM$_{2.5}$ at the four considered urban air quality measurement stations. The predicted time series have been presented only for the periods, for which the observations were available.

Annex C. The evaluation of the predicted trends of concentrations of PM$_{2.5}$ for longer periods, using
the measured concentrations of PM$_{10}$ and TSP as proxy variables.

The observed PM$_{10}$ and TSP (total suspended particles) concentrations were used to approximate the
PM$_{2.5}$ concentrations at five measurement stations (Kallio2, Mannerheimintie, Leppävaara4,
Tikkurila3 and Luukki). The use of the so-called scaled PM$_{2.5}$ concentrations, evaluated based on
proxy variables, makes it possible to evaluate the reliability of the modelled trends of PM$_{2.5}$
concentrations for a longer period, compared with comparing the predictions only with the observed
PM$_{2.5}$ concentrations.

**Materials and methods**

The measurements of PM$_{10}$ used here were conducted continuously for each hour, whereas the
measurements of TSP were conducted manually only as daily averages for every other day.

For each of the considered five stations, the scaled PM$_{2.5}$ concentration values were computed for
such years, for which the PM$_{2.5}$ concentration measurements had not yet been started, but either the
measured PM$_{10}$ or TSP concentration data, or both of those, were available at the reference station.
For instance, at the station of Mannerheimintie, the scaled PM$_{2.5}$ concentration values that were based
on the PM$_{10}$ and TSP measurements were therefore computed from 1997 to 2005, and from 1988 to
2005, respectively.

The so-called scaled PM$_{2.5}$ concentrations (denoted in the following as sPM2.5) were computed as
follows. At the station (i) during the year (j):

$$sPM2.5_i^j = f_H * f_i * rC_{rs}^{j,h_2}, j = 1, \dots, M \tag{C1}$$

where f$_H$ and f$_i$ are the scaling factors for the measurement height and for the station i, rC is the
measured concentration of a reference compound (in this study, PM$_{10}$ or TSP) at the height h$_2$ and rs
refers to a reference station. The subscript j refers to the years from 1 to M, i.e., from the earliest to
the most recent year, for which the concentration of the reference compound has been available,
before the start of the PM$_{2.5}$ measurements at the station i.

We have defined two criteria for the use of the measurements of a reference compound (PM$_{10}$ or
TSP). Such measurements can be used, if (i) the measured concentrations of a reference compound
at a reference station have been available for at least a couple of years simultaneously with the
measurements of the PM$_{2.5}$ concentrations at station i, and (ii) the measured concentrations of a
reference compound at a reference station have been available for at least a couple of years before
the measurements of the PM$_{2.5}$ concentrations were started at the station i. The station of Vallila in
central Helsinki was selected as a reference station for all of the considered stations, as that data
satisfied the above mentioned criteria, and the available datasets were the most extensive ones at that
station.

Scaling factor $f_i$ for station i is defined as the average of the annually averaged concentration ratios of $PM_{2.5}$ at station i and that of a reference compound at the reference station during N years, measured at the height $h_1$:

$$f_i = \frac{1}{N}\sum_{k=1}^{N}\left(\frac{PM2.5_i^{k,h_1}}{rC_{rs}^{k,h_1}}\right) \qquad (C2)$$

For computing these concentrations ratios, only simultaneously measured data is used that has been
available both at the reference station and at the station i. The averaging periods for computing these concentration ratios varied from 3 to 16 years, depending on the station and the reference compound ($PM_{10}$ or TSP). For each station and each reference compound, we then used an average value determined for the whole period, for which these concentrations were simultaneously available. This procedure is based on the assumption that the relative changes of these concentration ratios in time
are substantially smaller, compared with the relative changes of the $PM_{2.5}$ concentrations, over the considered time period.

It is recommended also to adjust the scaling, if the measurement height has substantially varied during the considered period. The scaling factor for the measurement height $f_H$ is defined as:

$$f_H = \frac{\frac{1}{P}\sum_{k=1}^{P} rC_{rs}^{h_1}}{\frac{1}{P}\sum_{k=1}^{P} rC_{rs}^{h_2}} \qquad (C3)$$

where rC are the reference concentrations at the reference station (rs) at the heights $h_1$ and $h_2$,. The $f_H$ factor is therefore simply the ratio of the average concentrations at the measurement heights during P years. However, the concentrations at heights h1 and h2 can be from different years. This can cause uncertainty to the height correction factor.

Two measurement heights were used for TSP at the station of Vallila, the height of 4 m during 2004-
2008 and the height of 12 m during 1988-2003. The yearly average concentrations of TSP for five years were used for both measurement heights, 4 m and 12 m. The $f_H$ value of 1.31 was used for all stations, for which the scaling by Vallila TSP data was possible (i.e. Kallio2, Mannerheimintie, and Luukki). In case of $PM_{10}$, the measurement height at Vallila did not differ significantly during the years (4 m and 4.5 m), and the height adjustment was not necessary, i.e., we used the value $f_H=1$. The
measurements of $PM_{2.5}$ were conducted at the heights of 4 m or 4.5 m, depending on the station.

**The results**

The periods, for which one can evaluate the accuracy of the modelled trends, were substantially longer (by a factor of from two to four) than for the case of using only the measured values of $PM_{2.5}$ for the
evaluation. In case of $PM_{10}$ and TSP, these values extend until 1997 and 1988, respectively. At two of the stations (Leppävaara and Tikkurila), there were no $PM_{2.5}$ measurements available at the same time as TSP data was measured at the reference station; for those stations we have therefore presented only the proxy values based on the $PM_{10}$ concentrations.

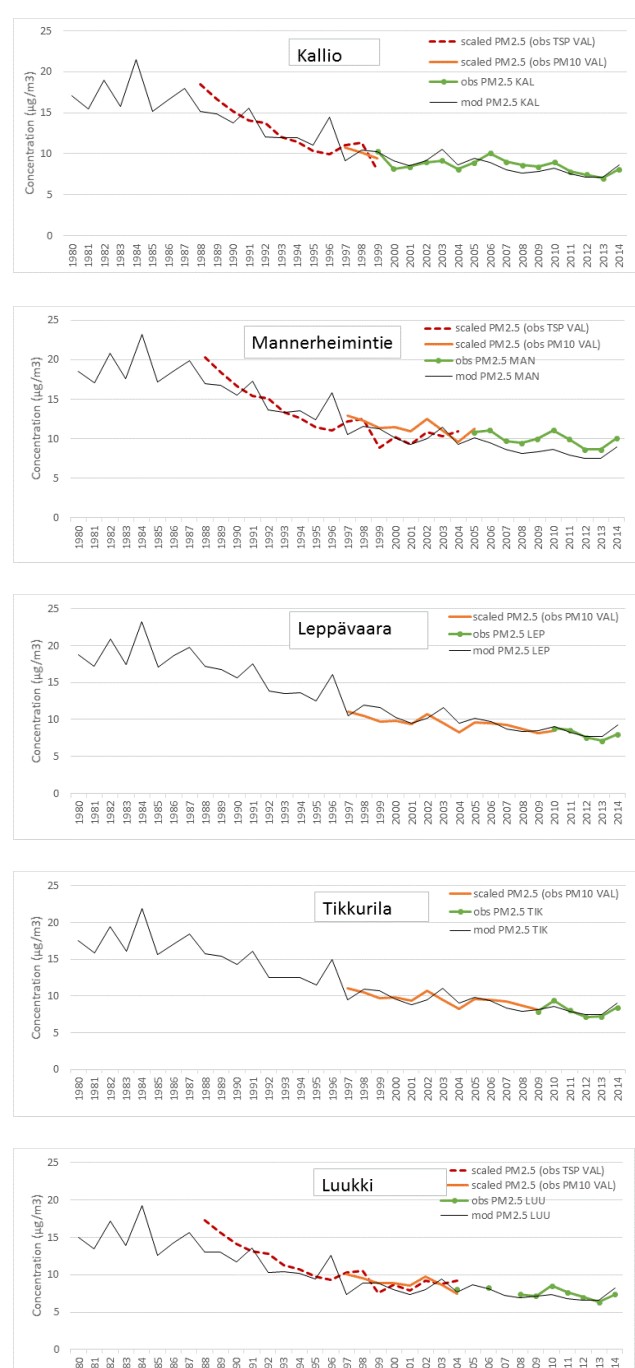



Fig C1 a-e. The modelled trends of concentrations of PM$_{2.5}$, compared with the measured concentrations of PM$_{2.5}$, and with the scaled concentrations, which were based on the measurements of the proxy variables PM$_{10}$ and TSP. The modelled PM$_{2.5}$ values have been presented by the black lines, the observed values of PM$_{2.5}$ with the green lines, and the scaled concentrations with the orange (based on PM$_{10}$) and dashed dark red (based on TSP) lines.


The trends of the concentrations, as evaluated using the scaled concentration values based on the measured PM$_{10}$ concentrations, agree well (all stations, except for Mannerheimintie) or fairly well

(Mannerheimintie) with the modelled long-term evolution of concentrations. This adds confidence that the modelled trends are fairly accurate, at least from the late 1990's to 2014.

The trends evaluated using TSP as a proxy variable also indicate a substantial decrease of the $PM_{2.5}$ concentrations since the late 1980's. Also these trends are qualitatively in agreement with the modelled trends of the $PM_{2.5}$ concentrations. However, the scaled concentrations are slightly higher for the earliest years considered, in the late 1980's, compared with the predicted concentrations.

Table 1. Selected statistical parameters associated with the agreement of the measured and predicted daily averaged concentrations of PM$_{2.5}$, at the regional background station of Luukki, from 2004 to 2014. The average values of the measurements and predictions, and the numbers of data points have also been presented.


| | Luukki (daily values) | 2004 | 2006 | 2008 | 2009 | 2010 | 2011 | 2012 | 2013 | 2014 |
|---|---|---|---|---|---|---|---|---|---|---|
| IA | Index of agreement | 0.67 | 0.65 | 0.73 | 0.72 | 0.62 | 0.65 | 0.68 | 0.60 | 0.66 |
| F2 | Factor-of-two | 61 | 59 | 54 | 66 | 61 | 55 | 64 | 58 | 61 |
| FB | Fractional bias | -0.06 | -0.03 | -0.06 | -0.01 | -0.14 | -0.11 | -0.04 | 0.03 | 0.11 |
| AvgCp | Predicted | 7.6 | 8.1 | 7.0 | 7.1 | 7.4 | 6.9 | 6.7 | 6.5 | 8.2 |
| AvgCo | Observed | 8.1 | 8.3 | 7.4 | 7.2 | 8.5 | 7.7 | 7.0 | 6.3 | 7.4 |
| N | Number of datapoints | 365 | 360 | 364 | 365 | 360 | 358 | 348 | 365 | 364 |

Table 2. Three selected statistical parameters associated with the agreement of the measured and predicted daily averaged concentrations of $PM_{2.5}$, at four urban measurement stations, from 1999 to 2014. The average values of the measurements and predictions, and the numbers of data points have also been presented.

| Kallio2 | | 1999 | 2000 | 2001 | 2002 | 2003 | 2004 | 2005 | 2006 | 2007 | 2008 | 2009 | 2010 | 2011 | 2012 | 2013 | 2014 |
|---|---|---|---|---|---|---|---|---|---|---|---|---|---|---|---|---|---|
| IA | Index of agreement | 0.75 | 0.70 | 0.72 | 0.71 | 0.71 | 0.66 | 0.81 | 0.69 | 0.75 | 0.77 | 0.75 | 0.65 | 0.75 | 0.69 | 0.64 | 0.71 |
| F2 | Factor-of-two | 80 | 81 | 78 | 72 | 71 | 70 | 73 | 64 | 62 | 56 | 73 | 67 | 63 | 66 | 66 | 65 |
| FB | Fractional bias | -0.01 | 0.12 | 0.01 | 0.02 | 0.14 | 0.06 | 0.07 | -0.12 | -0.12 | -0.11 | -0.07 | -0.08 | -0.03 | -0.06 | 0.01 | 0.07 |
| AvgCp | Average of predicted data | 10.2 | 9.2 | 8.5 | 9.2 | 10.6 | 8.6 | 9.6 | 9.0 | 8.0 | 7.7 | 7.9 | 8.3 | 7.5 | 7.0 | 7.1 | 8.7 |
| AvgCo | Average of observed data | 10.3 | 8.2 | 8.4 | 9.0 | 9.2 | 8.1 | 8.9 | 10.0 | 9.0 | 8.6 | 8.4 | 9.0 | 7.8 | 7.4 | 7.0 | 8.1 |
| N | Number of datapoints | 347 | 366 | 365 | 365 | 365 | 366 | 360 | 365 | 365 | 361 | 365 | 363 | 358 | 363 | 364 | 365 |
| **Mannerheimintie** | | | | | | | | 2005 | 2006 | 2007 | 2008 | 2009 | 2010 | 2011 | 2012 | 2013 | 2014 |
| IA | Index of agreement | | | | | | | 0.84 | 0.66 | 0.73 | 0.77 | 0.75 | 0.63 | 0.73 | 0.71 | 0.61 | 0.72 |
| F2 | Factor-of-two | | | | | | | 74 | 67 | 68 | 64 | 71 | 61 | 60 | 69 | 65 | 63 |
| FB | Fractional bias | | | | | | | -0.07 | -0.15 | -0.12 | -0.14 | -0.18 | -0.24 | -0.22 | -0.14 | -0.15 | -0.12 |
| AvgCp | Average of predicted data | | | | | | | 10.1 | 9.5 | 8.6 | 8.2 | 8.3 | 8.7 | 7.9 | 7.5 | 7.4 | 9.0 |
| AvgCo | Average of observed data | | | | | | | 10.8 | 11.1 | 9.7 | 9.4 | 10.0 | 11.0 | 9.9 | 8.6 | 8.6 | 10.1 |
| N | Number of datapoints | | | | | | | 360 | 365 | 365 | 365 | 365 | 365 | 363 | 356 | 363 | 364 |
| **Leppävaara4** | | | | | | | | | | | | | 2010 | 2011 | 2012 | 2013 | 2014 |
| IA | Index of agreement | | | | | | | | | | | | 0.75 | 0.77 | 0.70 | 0.70 | 0.71 |
| F2 | Factor-of-two | | | | | | | | | | | | 77 | 72 | 74 | 75 | 68 |
| FB | Fractional bias | | | | | | | | | | | | 0.03 | -0.04 | 0.02 | 0.08 | 0.15 |
| AvgCp | Average of predicted data | | | | | | | | | | | | 9.0 | 8.2 | 7.7 | 7.7 | 9.3 |
| AvgCo | Average of observed data | | | | | | | | | | | | 8.8 | 8.6 | 7.6 | 7.1 | 8.0 |
| N | Number of datapoints | | | | | | | | | | | | 355 | 364 | 366 | 363 | 363 |
| **Tikkurila3** | | | | | | | | | | | | 2009 | 2010 | 2011 | 2012 | 2013 | 2014 |
| IA | Index of agreement | | | | | | | | | | | 0.71 | 0.81 | 0.83 | 0.77 | 0.75 | 0.74 |
| F2 | Factor-of-two | | | | | | | | | | | 78 | 77 | 75 | 75 | 75 | 69 |
| FB | Fractional bias | | | | | | | | | | | 0.03 | -0.09 | -0.02 | 0.05 | 0.03 | 0.07 |
| AvgCp | Average of predicted data | | | | | | | | | | | 8.2 | 8.6 | 7.9 | 7.5 | 7.4 | 9.0 |
| AvgCo | Average of observed data | | | | | | | | | | | 7.9 | 9.4 | 8.0 | 7.1 | 7.2 | 8.4 |
| N | Number of datapoints | | | | | | | | | | | 365 | 365 | 365 | 366 | 363 | 365 |

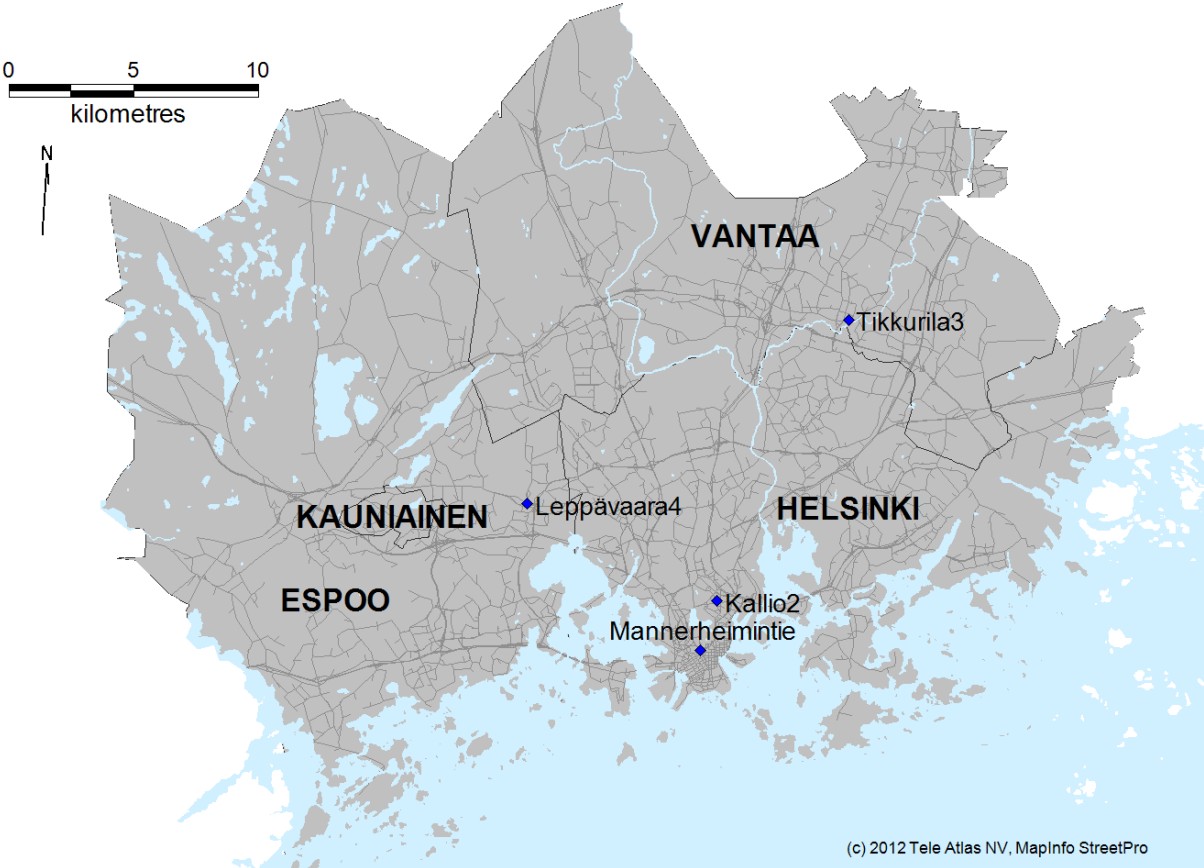


Fig. 1. The cities in the Helsinki Metropolitan Area (Helsinki, Espoo, Vantaa and Kauniainen), and the selected four measurement stations. The geographical extent of the area is approximately 35 x 30 km.

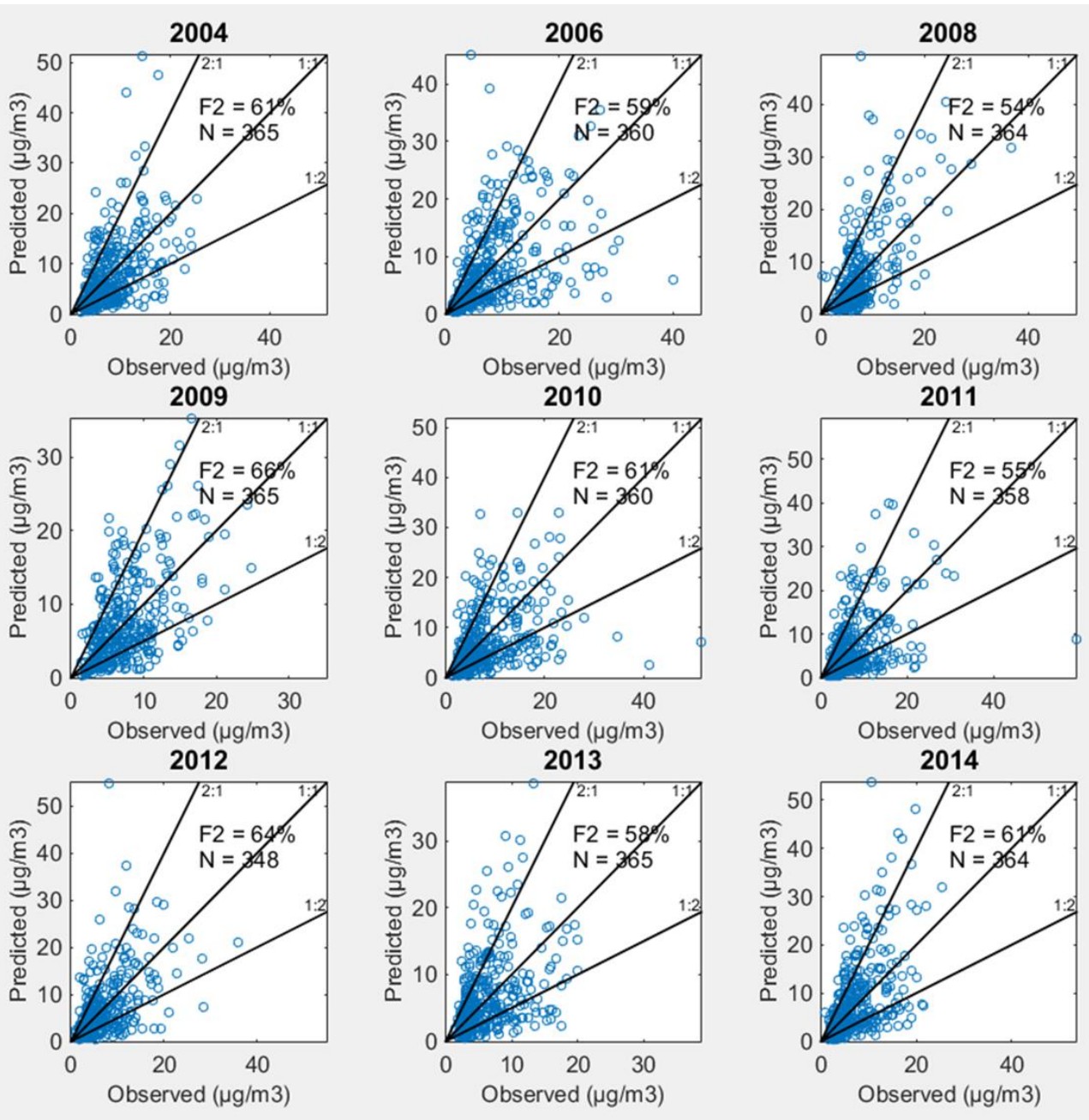

Figs. 2a-k. The scatter plots of measured and predicted daily average concentrations of $PM_{2.5}$ at the station of Luukki, for the period 2004 – 2014. The factor of two –values and the numbers of measured days have also been presented in the panels.





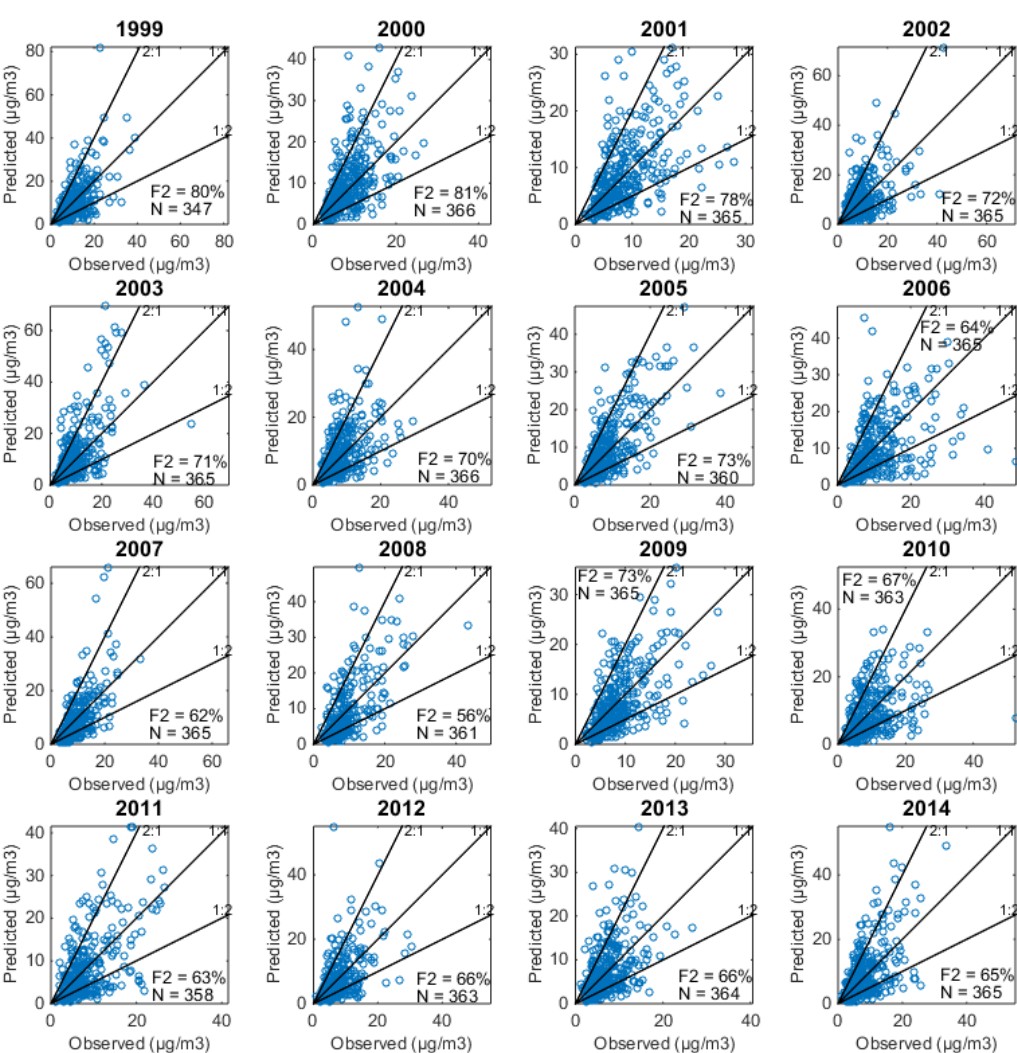

Figs. 3a-p. The scatter plots of measured and predicted daily average concentrations of $PM_{2.5}$ at the station of Kallio2, for the period 1999 – 2014. The factor of two –values and the numbers of measured days have also been presented in the panels.

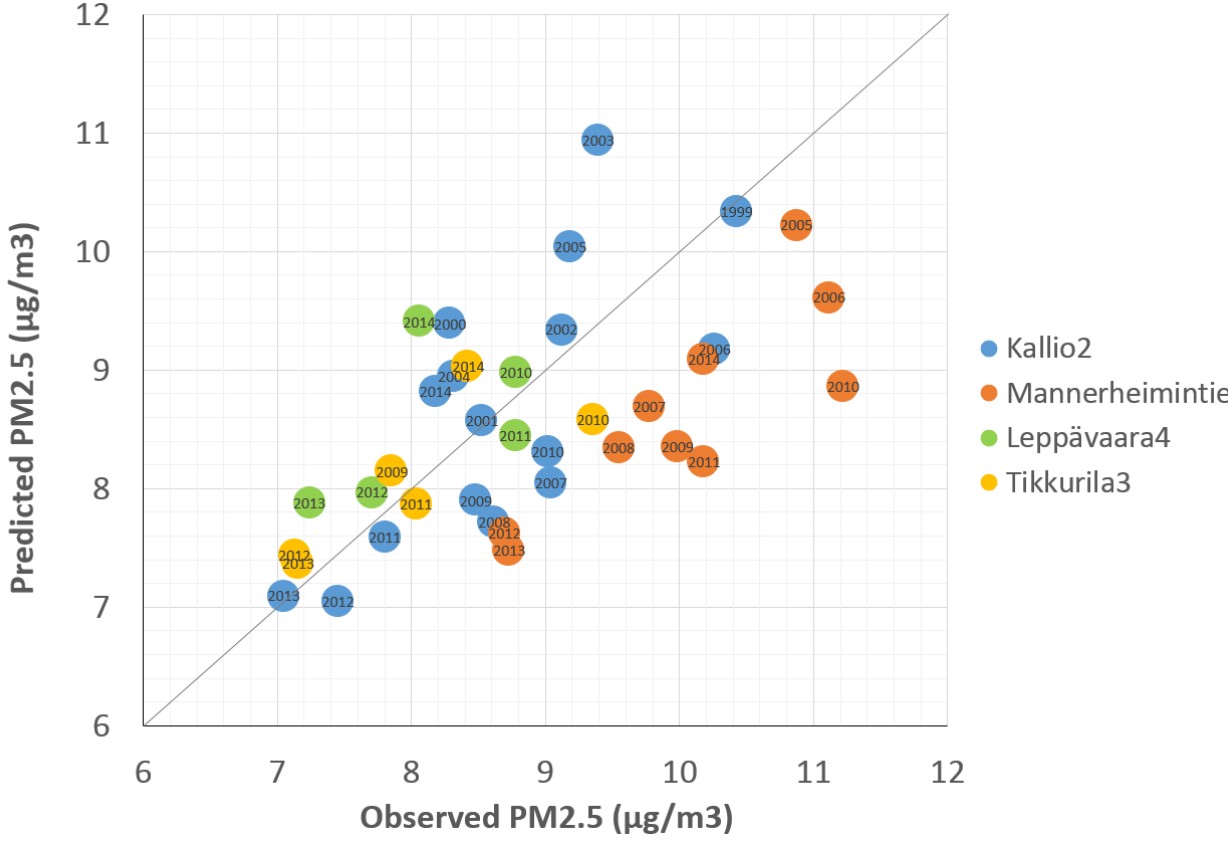

Fig. 4.The scatter plot of measured and predicted annual average concentrations of $PM_{2.5}$ at four measurement stations. The lowest values of both axes have been selected to be 6 µg/m$^3$.

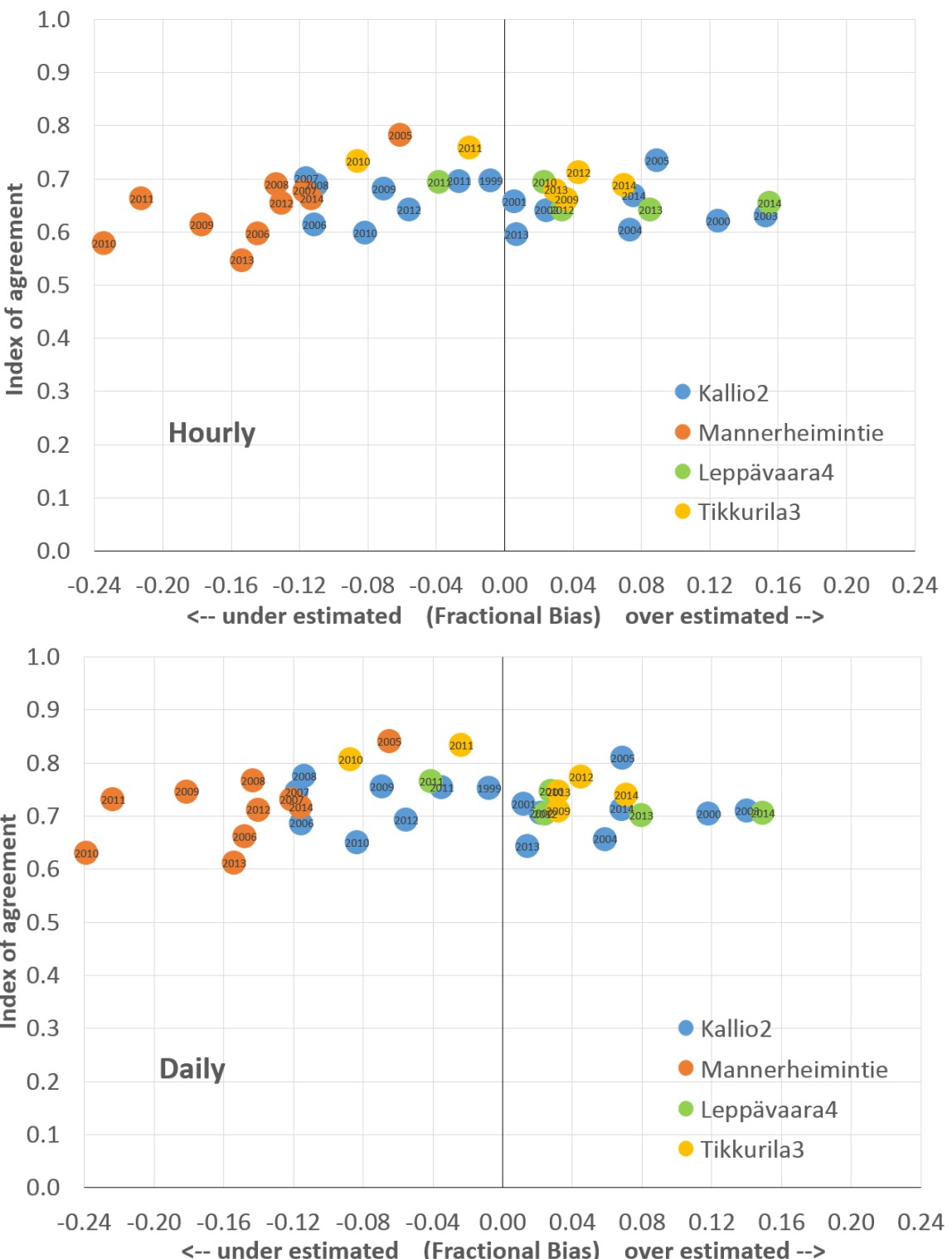

Figs. 5a-b. The annually evaluated indices of agreement and fractional biases, shown separately at four measurement stations and for each year, for hourly (upper panel) and daily (lower panel) concentrations of PM$_{2.5}$. The stations have been indicated by the various colours, and the years by small text inside each dot.


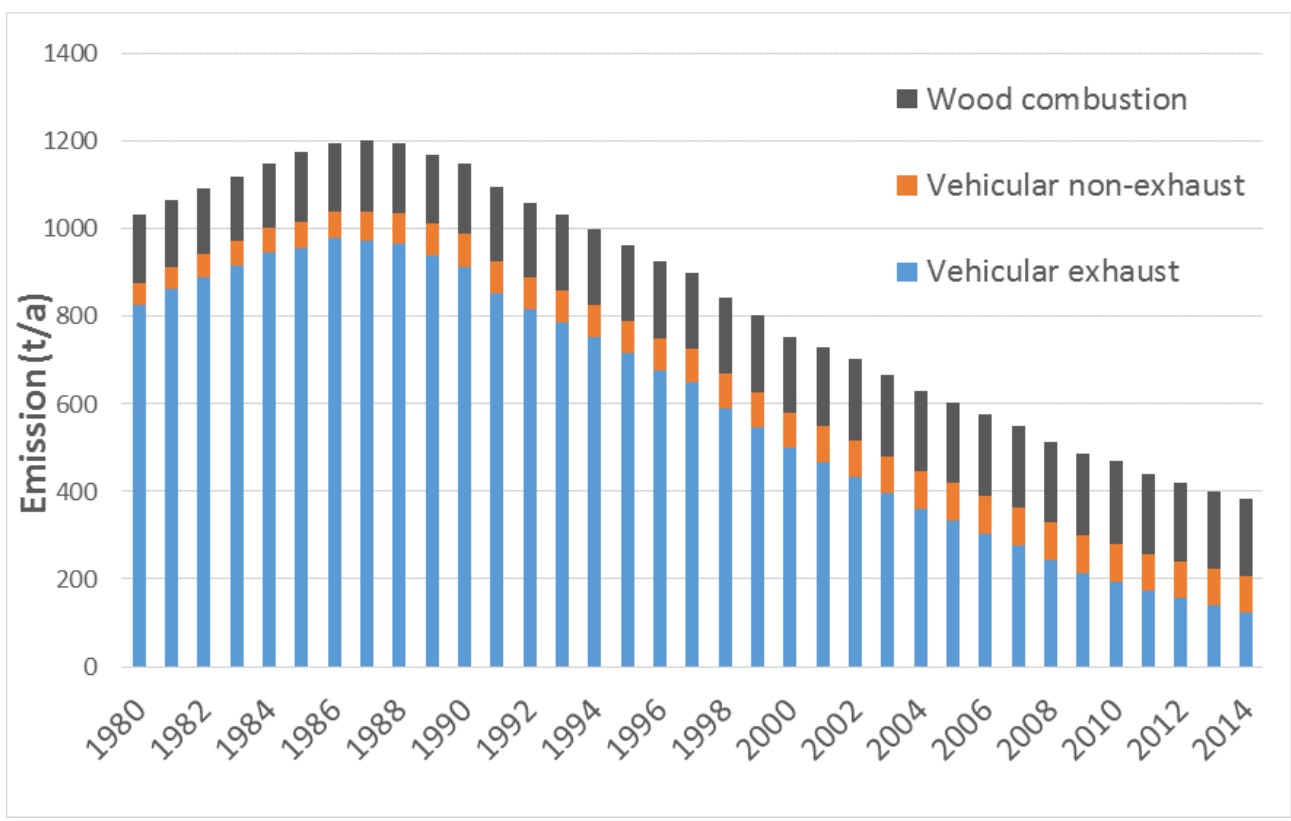

Fig. 6. The predicted emissions originated from vehicular sources and small-scale wood combustion in the Helsinki Metropolitan Area from 1980 to 2014. The vehicular emissions have been presented separately for exhaust and non-exhaust sources. The values for the vehicular exhaust emissions have been extracted from the LIPASTO system (Mäkelä and Auvinen, 2009).


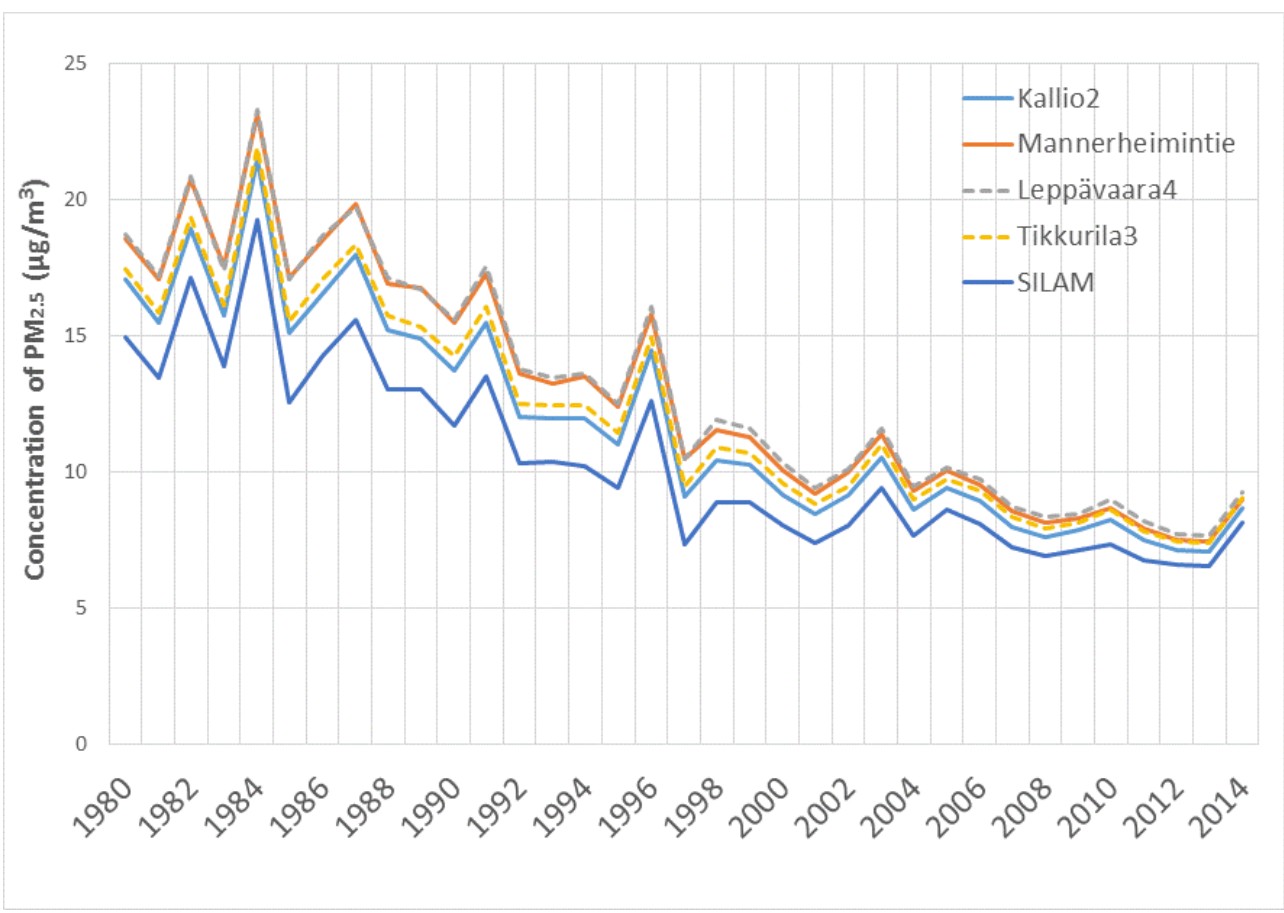

Fig. 7. The predicted annual average concentrations of PM$_{2.5}$ at four stations in the Helsinki
Metropolitan Area from 1980 to 2014, and the predicted regional background concentrations
(SILAM).

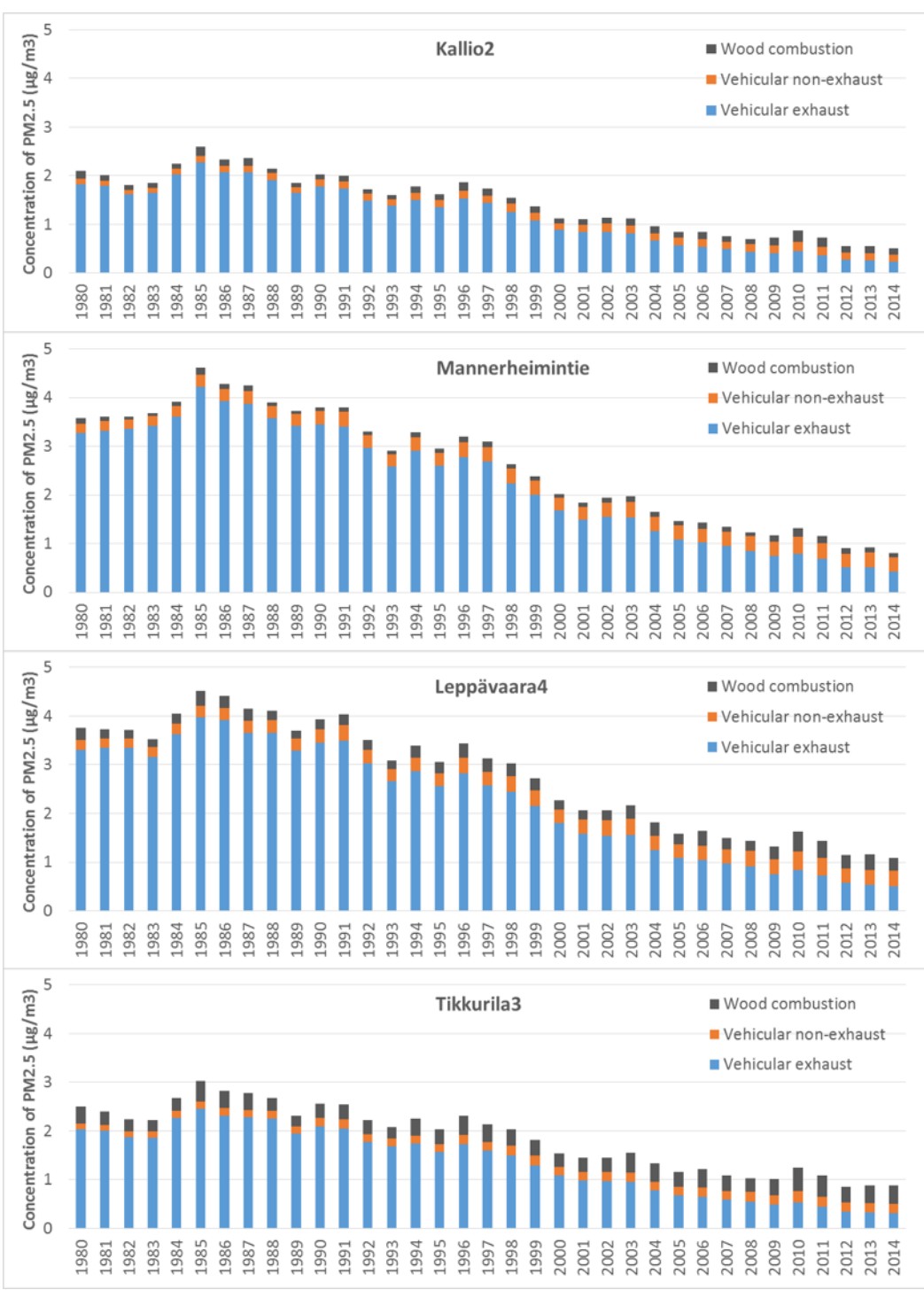


Figs. 8a-d. The predicted annual average concentrations of PM$_{2.5}$ originated from the three most important local source categories, i.e., small-scale combustion, vehicular non-exhaust and vehicular exhaust, at four measurement stations.