# Peer review of "Modelling of the urban concentrations of $PM_{2.5}$ on a high resolution for a period of 35 years, for the assessment of lifetime exposure and health effects"

_Atmospheric Chemistry and Physics, 2017_

## Referee Comment (RC1) · N. Moussiopoulos (Referee) · 20 Jan 2018

General comments

This paper discusses the long-term evolution of urban scale PM2.5 concentrations as input information for health impact assessment studies. The authors are very experienced in air pollution research and had key contributions to several important projects, as well as to significant previous publications.

The main novelty in this work in indeed the duration of the study period, which is indis-

pensable, if the air quality information is to be used for lifetime environmental exposure assessment. This requirement forced the authors to numerous assumptions, given that essential input information is lacking, mainly for the first decades of the study period. Apart from this novelty, this paper deserves been published in ACP, as it demonstrates that useful scientific results can be obtained following a pragmatic approach, i.e., adopting a simplified methodology and working with reasonable assumptions.

An important achievement of the authors is the quantification of the PM2.5 emission trends for both vehicular traffic and small-scale combustion, underlining the importance of the latter sector for future air pollution abatement.

Specific comments

In their updated manuscript version, the authors took into account several remarks communicated to them in the initial paper evaluation phase. Yet, there are still several aspects calling for further improvements of this paper prior to its publication:

(1) The updated version of the manuscript includes the statement that the contribution of shipping to the total PM2.5 concentrations can be higher than 20% within a distance of one kilometer from the harbor. At the same time it is stated that the target spatial resolution in this study is as high as 10 meters. In view of this obvious contradiction, the authors should consider shipping (and possibly also other) emissions in their UDM-FMI simulations, at least for a number of years (e.g., 2012-2014) to investigate the impact of these emission sources. It is likely that these additional calculations will allow them to justify convincingly their decision to concentrate on vehicular traffic and small-scale combustion.

(2) Two scaling assumptions are made in the overall analysis of vehicular traffic emissions: Firstly for traffic flow, using the mileage trend data (base year for the data: 2008), and secondly for exhaust emissions, using ratios of total exhaust emissions for different years (base year for the emission inventory: 2012). This reviewer accepts the authors' opinion that data are hardly available for the 80's and the 90's of the previous century,

but this is certainly not the case for the present decade. Therefore, the scaling results should be compared with information for both traffic volume data and the exhaust emissions for selected road links in more recent years than 2008 and 2012, respectively. By the way, in the manuscript it is mentioned that a substantial road reconstruction work was in progress during 2010-2011 in central Helsinki. Wouldn't it be a good idea to modify the scaling results to account for the consequences of this reconstruction?

(3) In their section 2.3.2 the authors discuss in detail the contributions of various heating sources and other fireplaces, adopting again a scaling procedure. Yet, what about advances in technology and fuel used? Given the technological improvements of the funaces applied, is it realistic to assume that the emission factors remained unchanged over the 35 years? Besides, several new fuel types (e.g., pellets) were introduced in this long period. These issues should be discussed in the paper.

(4) Regarding meteorology, the same parameters are used for the whole of the Helsinki Metropolitan Area covering approximately 1000 square kilometers. Perhaps it is unavoidable to work with this assumption in view of the overall pragmatic approach in this paper, yet the authors should discuss in quantitative terms the uncertainties possibly arising from this simplification.

Technical corrections

The manuscript is carefully written and well structured, and no language errors could be detected. So, no corrections of technical nature are necessary.

---

## Referee Comment (RC2) · Anonymous Referee #1 · 29 Jan 2018

The authors present a numerical modelling study of PM2.5 concentrations in the Helsinki area over the period 1980-2014. They use a multiple source gas dispersion model to simulate local primary emissions and a global model with suitable downscaling to estimate the regional background concentrations. Measured data from a regional background station over the period 1999 to 2014 and four urban measurement stations with a more limited dataset are used to evaluate predictions from the model.

The numerical models and approaches used in this work seem highly conventional and

it is hard to identify what aspects of the work are novel. The only aspect which might be considered novel in this context is the length of the period simulated and the use of techniques, albeit very straightforward, to estimate emissions from earlier time series for which inventory data are non-existent or not adequate. Since measured data are only available for less than half of the series simulated, it is not possible to evaluate whether the estimates of historical emissions are reliable.

The results show concentrations are dominated by the regional background, but the paper gives little detail of the model used, preferring to focus upon the local scale model which accounts for only a minor part of the PM2.5 concentrations. The omission of brake and tyre wear emissions (likely to exceed exhaust emissions in the latter years of the study) and cooking emissions and the use of an over-simplified parameterisation of resuspension emissions are also major weaknesses.

The performance of the model is far from good. The annual values for the square of correlation coefficients in Table 1 based upon measured and predicted daily average concentrations of PM2.5 at the regional background station range from 0.10 to 0.44. This clearly indicates that in many of the years the correlation coefficients were so low that the measured and predicted concentrations are almost uncorrelated. Table 2, which presents data for the other stations, does not include values of r2 and no reason for this is given. However, it is evident that a substantial proportion of the predicted concentrations are not within a factor of 2 of the measured concentrations, which again suggests that the simulations are not good, which is the conclusion drawn from Figure 2 and Figure 3 which show scatter plots of predicted versus observed concentrations. Even annual average concentrations in which much of the variability of the data is averaged out show considerable divergences between predicted and observed concentrations (Figure 4). It is likely that estimates for earlier years are even more uncertain.

In addition to these major issues which throw into doubt the value of publishing this study, the following points also require attention:

(a) Section 2.3.2 deals with emissions from small scale combustion. Clearly, some effort has been given to estimating activity data for earlier years but no indication is given of the emission factor used. The literature contains a very wide range of emission factors for wood combustion and this is a potentially important component of the model. Is there reason to believe that the emission factors have remained constant over the period of the study, given the changing technology of woodstoves?

(b) Section 3.1.2 describes the urban measurement stations but does not indicate the years for which data were available. It also indicates that selected results are presented in Table 2. Were these selected to give the best results, and if not, why were all data not included?

(c) The last paragraph of Section 3.1.2 does not make sense as the second sentence appears to be incomplete.
* * *

---

## Author Comment (AC1) · 25 Apr 2018

Dear editor, reviewers,

Please find attached a supplement that includes the final author comments and the revised manuscript.

In the revised manuscript, our suggested revisions are marked with red font.

Regards, Jaakko Kukkonen

[Figure]

Please also note the supplement to this comment:
https://www.atmos-chem-phys-discuss.net/acp-2017-968/acp-2017-968-AC1-
supplement.zip
* * *

---

## Author Response (AR1)

**Response to reviewers' comments, 25 April 2018**

For clarity, in the following the reviewers' comments have been indicated in blue font. Our response is in black font, and the changes made to the revised manuscript are written in red font.

**Anonymous Referee #1**

The authors present a numerical modelling study of PM2.5 concentrations in the Helsinki area over the period 1980-2014. They use a multiple source gas dispersion model to simulate local primary emissions and a global model with suitable downscaling to estimate the regional background concentrations. Measured data from a regional background station over the period 1999 to 2014 and four urban measurement stations with a more limited dataset are used to evaluate predictions from the model.

The numerical models and approaches used in this work seem highly conventional and it is hard to identify what aspects of the work are novel. The only aspect which might be considered novel in this context is the length of the period simulated and the use of techniques, albeit very straightforward, to estimate emissions from earlier time series for which inventory data are non-existent or not adequate.

We have revised the manuscript to show more clearly what parts of the modelling are new.

First, we have evaluated the global, regional and urban air quality for a multi-decadal period on a high resolution. Such multi-scale modelling has not been done previously internationally for any urban region.

The resulting modelled regional and global datasets will be made publicly available after their publication; a first version of these datasets is already available for collaborative research. These datasets could potentially be very valuable for researchers worldwide. E.g., suppose that one will need to make an assessment of air quality or a health impact assessment for any city or fairly limited domain globally, for any time period from 1980 to 2014. The regional scale computations of this study could then be used as boundary conditions, simply by downloading these data from the web. This could in many cases save a huge amount of resources, compared with the commonly used option, i.e., starting such very laborious larger scale emission and dispersion computations from scratch.

We have added a comment on these possibilities to the last paragraph of the conclusions.

Second, the urban scale assessments also include many novel inventories and sub-models. The emission inventory of small-scale combustion is totally new for this area. This inventory is also very detailed and comprehensive, compared with other corresponding inventories published elsewhere. The structure and application of this inventory for $PM_{2.5}$ has been presented for the first time in this manuscript. The description of how exactly the inventory was made could also be valuable for the design of other similar inventories elsewhere.

We have added one sentence to the last paragraph of the introduction to make this clearer. We have also substantially improved the description of this inventory in the revised manuscript, and its extension for a longer period.

Regarding the urban scale assessments, it is also novel to extend the urban emission inventories regarding vehicular traffic (exhausts and suspension) and small-scale combustion for a multi-decadal period. We have developed procedures for such extensions, using the best available data and materials. This has not been a trivial task; clearly, the input data is scarcer and less reliable for the earlier years. These procedures could also potentially be useful for making such projections for historical years elsewhere.

We have added one sentence to the last paragraph of the introduction to make this clearer.

Third, all the long-term numerical results on both emissions and concentrations are totally new. Such multi-decadal assessments of air quality have not been previously published for any city in the reviewed literature. These long-term results and trends were also thoroughly evaluated against the measured data.

Since measured data are only available for less than half of the series simulated, it is not possible to evaluate whether the estimates of historical emissions are reliable.

In response to the reviewer's comment, we have included into the revised manuscript an evaluation for substantially longer time periods (Annex C, and some text in the section on trends, 3.2., and some comments in the conclusions section). This can be done indirectly, using so-called proxy analyses.

We evaluated the accuracy of the modelled trends of concentrations of $PM_{2.5}$, by using the measured values of the concentrations of $PM_{10}$ and TSP (total suspended particles) as proxy variables. Using such an analysis, the model results can be evaluated for a substantially longer period than would be possible using only the $PM_{2.5}$ measurements. In particular, the trends of the concentrations using $PM_{10}$ as a proxy agreed well at four stations, and fairly well at one station, with the modelled long-term evolution of concentrations. This adds more confidence that the modelled trends are fairly accurate.

The results show concentrations are dominated by the regional background, but the paper gives little detail of the model used, preferring to focus upon the local scale model which accounts for only a minor part of the PM2.5 concentrations.

The regional background indeed constitutes most of the $PM_{2.5}$ concentrations, so we have therefore added, as per reviewer request, more discussion on the regional modelling to the revised manuscript. Especially, the treatments for non-anthropogenic emissions have been described better, and we have included more information and more of the most relevant references on the various modelling choices.

However, the main focus of the present paper is not intended to be on regional and global computations. Writing of a separate manuscript is in progress on the global and regional computations; some preliminary results of these were already presented by Sofiev M. et al. (2018) (A Long-Term Re-Analysis of Atmospheric Composition and Air Quality. In: Mensink C., Kallos G. (eds) Air Pollution Modeling and its Application XXV. ITM 2016. Springer Proceedings in Complexity. Springer, Cham).

The first aim of this study was to provide an accurate and reliable high-resolution database on the urban scale concentrations of $PM_{2.5}$, to be used in the subsequent health impact assessments (as stated in the introduction). For health impact assessment, also the urban-scale spatial differences and short-term (hourly) variations of air quality are important. The local sources are responsible for most of such fine-resolution temporal differences.

The omission of brake and tyre wear emissions (likely to exceed exhaust emissions in the latter years of the study) and cooking emissions and the use of an over-simplified parameterization of resuspension emissions are also major weaknesses.

The emissions from brake, tyre and clutch wear (BTW) were partly, but not completely included. However, we evaluated their relative influence based on another study for this region, Kupiainen et al. (2015), in section "Evaluation of the emissions of suspended dust for the target period". We have written more clearly, what exactly was evaluated, and what was not evaluated in the revised manuscript. We have also estimated quantitatively, how large the fraction that was not evaluated is; this has also been presented in the revised manuscript.

We have allowed for a vast majority of the vehicular non-exhaust emissions. The neglected fraction of vehicular non-exhaust emissions (compared with all vehicular non-exhaust emissions) was estimated to be smaller than 20 %. Vehicular suspension (i.e., non-exhaust) emissions were nearly as large as vehicular exhaust emissions for the most recent years for $PM_{2.5}$ (cf. Fig. 6).

It is not possible to use complex suspension models (such as NORTRIP or FORE) for so many years in the past, as the required sanding data was not available. This has been mentioned in the manuscript in the beginning of the section "Evaluation of the emissions of suspended dust …". We have therefore developed and used a simpler semi-empirical modelling approach.

Cooking emissions were taken into account, as wood-burning stoves, ovens and baking ovens were included in the emission inventory (Kaski et al, Table 14). However, the influence of cooking emissions has been evaluated to be fairly small for this region. Mostly, electric ovens are used for cooking. During the summer season, there is also some barbecue cooking outdoors, using charcoal, LNG or electric cooking, but its influence on air quality is negligible. Clearly, cooking can be much more important for air quality, e.g., in some southern European cities.

We have revised the discussion in the section 'Evaluation of the emissions of suspended dust …' to be clearer. A couple of sentences were added to the first paragraph of this section, to address first the basic ideas and approach.

As per reviewer request, we evaluated numerically the inaccuracies of the used simplified suspension modelling, compared with the predictions of the more complex model, for a few additional years.

The results of this analysis and their discussion have been included in the revised manuscript. The differences in annual concentration values from suspension, computed by the two models, was less than 30 % for all the three years considered.

The performance of the model is far from good. The annual values for the square of correlation coefficients in Table 1 based upon measured and predicted daily average concentrations of PM2.5 at the regional background station range from 0.10 to 0.44. This clearly indicates that in many of the years the correlation coefficients were so low that the measured and predicted concentrations are almost uncorrelated. Table 2, which presents data for the other stations, does not include values of r2 and no reason for this is given.

However, it is evident that a substantial proportion of the predicted concentrations are not within a factor of 2 of the measured concentrations, which again suggests that the simulations are not good, which is the conclusion drawn from Figure 2 and Figure 3 which show scatter plots of predicted versus observed concentrations. Even annual average concentrations in which much of the variability of the data is averaged out show considerable divergences between predicted and observed concentrations (Figure 4). It is likely that estimates for earlier years are even more uncertain.

The reviewer is correct in noting that R2 was included in Table 1 (for the regional background station of Luukki) but not in the subsequent corresponding tables (for the urban stations). These tables already contain two variables that describe the temporal correlation of the measured and predicted time series, viz. IA and F2, so presenting still another parameter for that aspect is not necessary. For consistency, we have therefore removed the R2 values from Table 1 in the revised manuscript.

We also checked again the computations for the station of Luukki, and found one slight mistake in the computations: the leap day was not correctly taken into account in the time series. However, we have

now corrected the results in Table 1 and Figs. 2 in that respect; the new values are slightly better than the previous ones.

We have also added an Annex B that presents the time series of the observed and predicted annual average concentrations of $PM_{2.5}$. These data will show more clearly the actual correlation for the annual average values, as these include no post-processing after the dispersion computations.

As stated in the paper, Luukki is a regional background station for the Helsinki region, in which the concentrations are dominated by the regional-scale pollution. In the current setup, this is represented by the computations of the SILAM model on a European scale, with a resolution of 0.5 degree. For a coastal city such as Helsinki, such a resolution can be too coarse for achieving a high correlation. However, a fairly good or good agreement of the long-term mean (e.g. annual) values can be achieved, and were actually achieved in this study. A corresponding clarification was added to the section "Regional background concentrations at one station".

In general, we felt that a reasonable way to assess how good or bad the model vs. measurements agreement was, would be to compare the model performance measures with similar studies found in the literature. We therefore included a brief discussion on a comparison with a similar study conducted for one year in London (Singh et al., 2014) at the end of the section 3.1. The results of the present study were slightly worse, but comparable, and these differences could be explained by the use of the semi-empirical modelling of the regional background in the Singh et al study, which is somewhat more accurate than modelling solely with any regional dispersion model. In the present study, the regional background was modelled with the chemical transport model SILAM, without any experimental adjustments.

Singh, Vikas, Ranjeet Sokhi and Jaakko Kukkonen, 2014. PM2.5 concentrations in London for 2008 - A modeling analysis of contributions from road traffic. Journal of the Air & Waste Management Association, 64:5, 509-518. DOI: 10.1080/10962247.2013.848244. http://dx.doi.org/10.1080/10962247.2013.848244.

The model performance measures that are based on correlations of measured and predicted data (such as, e.g., IA and $R^2$) are currently widely used in model evaluations. However, one could also criticize this practice. The computation of these correlations requires that the distributions of the concentrations need to be normal, which is not always the case in urban conditions. The model bias can actually therefore be the more important parameter with respect to the long-term health assessments.

In addition to these major issues which throw into doubt the value of publishing this study, the following points also require attention:

(a) Section 2.3.2 deals with emissions from small scale combustion. Clearly, some effort has been given to estimating activity data for earlier years but no indication is given of the emission factor used. The literature contains a very wide range of emission factors for wood combustion and this is a potentially important component of the model. Is there reason to believe that the emission factors have remained constant over the period of the study, given the changing technology of woodstoves?

We have substantially revised the section 2.3.2 according to the comments of the reviewer. After the required revisions, the section was fairly long. We have therefore also inserted subtitles, for a better readability of this section.

The emission factors have been properly reported in the revised manuscript, in a sub-section. We have also made clear, how exactly we have allowed for the temporal long-term changes of the total emissions from wood combustion. In particular, we added two new paragraphs in the beginning of that sub-section that describe, which factors we considered to be the most important ones, which were allowed for, and which we had to omit.

First, it should be noted that for estimating the temporal evolution of wood combustion, one needs to know several temporal developments, such as the numbers of detached and semidetached houses, the amounts of firewood used, the shares of primary heating sources, and the numbers of boilers and sauna stoves. The evolution of all of these factors have been evaluated in this study for the whole of the study period, based on the best available information (as explained in the revised manuscript).

We have also allowed for the changes of the shares of the various categories of heating devices. The general trend has been towards cleaner heating devices. In that respect, the most important improvements of technology (which is the increased usage of cleaner device categories) have been taken into account. However, there was not enough quantitative information to allow for the influence of technological changes, separately within each heating device category.

 (b) Section 3.1.2 describes the urban measurement stations but does not indicate the years for which data were available. It also indicates that selected results are presented in Table 2. Were these selected to give the best results, and if not, why were all data not included?

A comment has been added in the beginning of the section 3.1.2 to make it more clear, which data has been used. We also removed the term 'selected' from these contexts.

We have considered all the measurement data that have been available for the five measurement stations considered. For the stations of Kallio, Luukki, Mannerheimintie, Tikkurila and Leppävaara, the measured concentrations of $PM_{2.5}$ were available since 1999, 2004, 2005, 2009 and 2010, respectively.

'Selected results' referred only to selected statistical parameters. All the measured values (and model performance measures) have been presented in the tables, irrespective of how good the results were. The selection of statistical parameters were standard choices, not selected in any way to make the results look better.

 (c) The last paragraph of Section 3.1.2 does not make sense as the second sentence appears to be incomplete.

This sentence has been corrected in the revised manuscript.

**N. Moussiopoulos (Referee)**

General comments

This paper discusses the long-term evolution of urban scale PM2.5 concentrations as input information for health impact assessment studies. The authors are very experienced in air pollution research and had key contributions to several important projects, as well as to significant previous publications.

The main novelty in this work in indeed the duration of the study period, which is indispensable, if the air quality information is to be used for lifetime environmental exposure assessment. This requirement forced the authors to numerous assumptions, given that essential input information is lacking, mainly for the first decades of the study period. Apart from this novelty, this paper deserves been published in ACP, as it demonstrates that useful scientific results can be obtained following a pragmatic approach, i.e., adopting a simplified methodology and working with reasonable assumptions.

An important achievement of the authors is the quantification of the PM2.5 emission trends for both vehicular traffic and small-scale combustion, underlining the importance of the latter sector for future air pollution abatement.

Thank you.

Specific comments

In their updated manuscript version, the authors took into account several remarks communicated to them in the initial paper evaluation phase. Yet, there are still several aspects calling for further improvements of this paper prior to its publication:

(1) The updated version of the manuscript includes the statement that the contribution of shipping to the total PM2.5 concentrations can be higher than 20% within a distance of one kilometer from the harbor. At the same time it is stated that the target spatial resolution in this study is as high as 10 meters. In view of this obvious contradiction, …

We have written the result from Soares et al more clearly in the revised manuscript, in section 2.1.

The 20 % result was found in an earlier study by partly the same authors, Soares et al., 2014. Both in the computations of the present study, and in those by Soares et al., 2014, the computational resolution was much finer than one kilometer. However, considering the spatial distribution of the predicted shipping contributions, the values of 20 % at a certain distance (about one kilometer) from the major harbours were found.

… the authors should consider shipping (and possibly also other) emissions in their UDM-FMI simulations, at least for a number of years (e.g., 2012-2014) to investigate the impact of these emission sources. It is likely that these additional calculations will allow them to justify convincingly their decision to concentrate on vehicular traffic and small-scale combustion.

We have followed the reviewer's advice, and added new up-to-date results on the contribution of shipping, for a recent three-year period, 2012-2014. In order to keep the flow of text readable, we inserted these results to Annex A.

(2) Two scaling assumptions are made in the overall analysis of vehicular traffic emissions: Firstly for traffic flow, using the mileage trend data (base year for the data: 2008), and secondly for exhaust emissions, using ratios of total exhaust emissions for different years (base year for the emission inventory: 2012). This reviewer accepts the authors' opinion that data are hardly available for the 80's and the 90's of the previous century, but this is certainly not the case for the present decade. Therefore, the scaling results should be compared with information for both traffic volume data and the exhaust emissions for selected road links in more recent years than 2008 and 2012, respectively.

We have clarified and substantially rewritten the section on the "Evaluation of traffic flows and vehicular emissions for the whole target period".

The annual values of the total traffic flows and exhaust emissions from vehicular traffic in the Helsinki Metropolitan Area were extracted directly from the LIPASTO system. These values are available in the LIPASTO system, since the early 1980's. These are based on various measured data, such as vehicle fleet composition, measured traffic flows, emission measurements, etc.; we have described this more clearly in the revised manuscript. There is therefore no need to scale these values. We have only scaled the spatial distributions of the emissions for a longer period, within the road and street network in the area.

By the way, in the manuscript it is mentioned that a substantial road reconstruction work was in progress during 2010-2011 in central Helsinki. Wouldn't it be a good idea to modify the scaling results to account for the consequences of this reconstruction?

We have revised this statement to be more accurate and specific.

This mentioned (road) reconstruction was in progress only in a very limited area, along the Mannerheim street; it did not therefore affect the whole of the central area of Helsinki (and certainly not the whole of the considered area). It essentially affected only the concentration values of this specific station, for some time during the two-year period. The magnitude of the effect is also not known, so its quantification is not possible.

(3) In their section 2.3.2 the authors discuss in detail the contributions of various heating sources and other fireplaces, adopting again a scaling procedure. Yet, what about advances in technology and fuel used? Given the technological improvements of the funaces applied, is it realistic to assume that the emission factors remained unchanged over the 35 years? Besides, several new fuel types (e.g., pellets) were introduced in this long period. These issues should be discussed in the paper.

We have substantially revised this section. Please see our response to a very similar comment by reviewer 1.

In this area, the small-scale combustion that is not using oil or gas uses almost solely wood as a fuel. The share of pellet use is negligible in small-scale combustion.

(4) Regarding meteorology, the same parameters are used for the whole of the Helsinki Metropolitan Area covering approximately 1000 square kilometers. Perhaps it is unavoidable to work with this assumption in view of the overall pragmatic approach in this paper, yet the authors should discuss in quantitative terms the uncertainties possibly arising from this simplification.

The reviewer is correct in writing that it is not possible to allow for the spatial variations of meteorology within the selected area in this study.

The most important meteorological parameters for atmospheric diffusion are wind speed and atmospheric stability (e.g., Backman et al., 2017, reference below); in addition, ambient temperature and solar radiation are key parameters for the chemical transformation processes. The spatial and temporal variation of the urban-scale meteorological parameters in the Helsinki region have been studied by Wood et al. (2013a, b, below). However, these studies do not provide for any concrete methods for allowing for the spatial variation of the meteorological factors within this area.

Backman, John, Wood, Curtis R., Auvinen, Mikko, Kangas, Leena, Karppinen, Ari and Kukkonen, Jaakko, 2017. Sensitivity analysis of the meteorological pre-processor MPP-FMI 3.0 using algorithmic differentiation. Geosci. Model Dev., 10, 3793–3803, https://doi.org/10.5194/gmd-10-3793-2017

Wood, Curtis R; Leena Järvi; Rostislav D Kouznetsov; Annika Nordbo; Sylvain Joffre; Achim Drebs; Timo Vihma; Anne Hirsikko; Irene Suomi; Carl Fortelius; Ewan J O'Connor; Dmitri Moisseev; Sami Haapanala; Joonas Moilanen; Markku Kangas; Ari Karppinen; Timo Vesala; Jaakko Kukkonen, 2013a. An overview on the Urban Boundary-layer Atmosphere Network in Helsinki, Bulletin of the American Meteorological Society, 10.1175/BAMS-D-12-00146.1 http://dx.doi.org/10.1175/BAMS-D-12-00146.1

Wood CR, Kouznetsov RD, Gierens R, Nordbo A, Järvi L, Kallistratova MA, Kukkonen J, 2013b. On the temperature structure parameter and sensible heat flux over Helsinki from sonic anemometry and scintillometry. Journal of Atmospheric and Oceanic Technology, Vol 30, pp. 1604-1615.

The heat island effect in this region amounted only to 1 °C or less, on an annual average level, as found by Drebs (2011). (Helsinki urban heat island as temporal and spatial phenomena, Faculty of Science, University of Helsinki, Master thesis on physical geography). The coastal effects on meteorology, and

the topographical urban effects on wind flows and stability are therefore probably more important that the direct effects of the ambient temperature differences.

Unfortunately, we do not have a rigorous quantitative estimate on the influence of the variation of meteorological parameters in this area. However, our expert judgment is that this effect is clearly smaller than the influence of the uncertainties in evaluating the emissions, regionally and on an urban scale.

We added a semi-quantitative estimate in the discussion in the section 3.1.2, "Regional background concentrations".

Technical corrections

The manuscript is carefully written and well structured, and no language errors could be detected. So, no corrections of technical nature are necessary.

We have also done some slight technical editing of some of the figures, to show part of the texts on larger font, and some legends were revised to include more accurate descriptions.